# Skillful multiyear prediction of marine habitat shifts jointly constrained by ocean temperature and dissolved oxygen

Zhuomin Chen [1] ✉, Samantha Siedlecki[1], Matthew Long [2], Colleen M. Petrik [3], Charles A. Stock [4] & Curtis A. Deutsch [5]

The ability to anticipate marine habitat shifts responding to climate variability has high scientific and socioeconomic value. Here we quantify interannual-to-decadal predictability of habitat shifts by combining trait-based aerobic habitat constraints with a suite of initialized retrospective Earth System Model forecasts, for diverse marine ecotypes in the North American Large Marine Ecosystems. We find that aerobic habitat viability, defined by joint constraints of temperature and oxygen on organismal energy balance, is potentially predictable in the upper-600 m ocean, showing a substantial improvement over a simple persistence forecast. The skillful multiyear predictability is dominated by the oxygen component in most ecosystems, yielding higher predictability than previously estimated based on temperature alone. Notable predictability differences exist among ecotypes differing in temperature sensitivity of hypoxia vulnerability, especially along the northeast coast with predictability timescale ranging from 2 to 10 years. This tool will be critical in predicting marine habitat shifts in face of a changing climate.

Habitat shifts of marine species are closely associated with changes in ocean temperature ($T$) and dissolved oxygen ($O_2$) concentrations, as they are two fundamental constraints on aerobic metabolism of marine species and consequently determine the viability of marine habitats[1–6]. Physically, warming of the oceans decreases $O_2$ levels directly as its solubility decreases in warmer waters, and indirectly as it intensifies upper ocean stratification, reducing ocean ventilation - thus suppressing the supply of oxygen into the ocean interior[5,7,8]. On the other hand, metabolic rates of organisms are $T$-dependent and increase exponentially with rising temperatures - thus leading to increased aerobic demand for $O_2$ (e.g., ref. 9). For marine habitats to be metabolically viable, the environmental supply of $O_2$ must meet not only the minimum physiological requirements for survival but also additional requirements for essential ecological activities of marine species such as feeding, defense, and reproduction[3]. This implies that

$O_2$ concentrations may be inadequate to meet the demand of species for oxygen under warmer conditions, probably resulting in geographical or vertical viable habitats shrinking or shifting. Because $O_2$ and $T$ exhibit strong decadal variability, predictions of the corresponding marine habitat shifts are potentially possible on multi-annual time scales.

A physiologically mechanistic framework has been developed recently that enables quantifying habitat constraints for different species arising from the metabolic dependence on $T$ and requirements for $O_2$[3,10]. Under this framework, an index termed the Metabolic Index ($\Phi$) is defined as the ratio of $O_2$ supply to an organism's resting metabolic demand, integrating consideration of both $O_2$ availability (in the form of partial $O_2$ pressure: $p_{O_2}$) and $T$ effects ("Methods" section)[3,10]. The Metabolic Index depends on two key metabolic traits of marine species: the hypoxic tolerance ($A_o$) and its temperature

[1]University of Connecticut, Department of Marine Sciences, Groton, CT 06340, USA. [2]Climate & Global Dynamics Laboratory, National Center for Atmospheric Research, Boulder, CO 80305, USA. [3]Scripps Institution of Oceanography, University of California San Diego, La Jolla, CA 92037, USA. [4]Geophysical Fluid Dynamics Laboratory, NOAA, Princeton University, Princeton, NJ 08540, USA. [5]Department of Geosciences/High Meadows Environmental Institute, Princeton University, Princeton, NJ 08540, USA. ✉e-mail: zhuomin.chen@uconn.edu

sensitivity ($E_o$). These traits have been synchronously determined for a diversity of species using published laboratory and field data[3]. Observed marine species distributions better coincide with the spatial distribution of this index than models based on either $T$ or $O_2$ alone[3,10]. Occurrences of marine species are only found where $\Phi$ is above a critical value ($\Phi_{crit}$), which equals to 1 for resting metabolism and is larger than 1 for active metabolism in support of growth and essential ecological activities[3,10]. The capacity of the $\Phi$ to identify viable habitats of marine species has been successfully employed to explore how variability in the environmental variables can affect the distribution of viable habitats for the past, contemporary, and projected future oceans[3,5,6,11]. In regions of strong climatic variability of $O_2$, its contribution to $\Phi$ is critical to explain observed variations in population abundance and geographic range[12,13], suggesting that $O_2$ may greatly enhance the predictability of marine habitat shifts.

Global Earth System Models (ESMs) have demonstrated skill in forecasting physical and biogeochemical variables important to marine species on seasonal to decadal time scales[14,15]. The predictive capabilities of ESMs in combination with the $\Phi$ enables the development of forecasts that explicitly consider impacts of $T$ and $O_2$ availability on habitat viability. This index in the upper-400 m ocean is projected to reduce by ~20% globally and by ~50% in northern high-latitude regions by the end of this century, due to the combined effects of warming and deoxygenation under a high emissions scenario[3]. The projected reductions would generate spatial shifts and contractions of habitats and habitable seasons, posing a significant risk for marine ecosystems[16–19]. Global and local extinction risks for various marine biota are assessed based on this index and compared across a range of emission scenarios over the next few hundreds of years[20]. While the long-term trends of projections are worrisome, there is likely important variability on the interannual-to-decadal timescale, and predictability of this index and its mechanistic controls on this time scale from the physical and biogeochemical driver variables are still unknown. This time horizon, however, is critical for marine resource management to reduce impacts, promote resilience, and maximize the value of living marine resources in the face of changing ocean conditions[14,21].

We focus on the normalized Metabolic Index ($\phi = \Phi/\Phi_{crit}$) in this work for a consistent critical value of the $\phi$ ($\phi_{crit} = 1$) for resting and active metabolisms. Like $\Phi$, $\phi$ is characterized for a species by two measurable traits. In this case, $A_c$ equals to $A_o/\Phi_{crit}$, which is the hypoxic tolerance normalized to its minimum value needed to sustain ecological activity. Thus, ecological habitability in a natural environment is defined by $\phi > 1$, in the same way that physiological habitability is defined by $\Phi > 1$. The two metabolic traits of marine species exhibit approximately normal distributions with the $A_c$ and $E_o$ mainly within the ranges of 0 to 20 atm$^{-1}$ and −0.2 to 1.0 eV respectively, based on the available trait database (Supplementary Fig. 1).

We examine the predictability of the $\phi$ on the interannual-to-decadal timescale in two depth habitats (0-200 m, the surface layer or epipelagic zone where most of the visible lights exist, and 200-600 m, the thermocline layer within the mesopelagic zone or twilight zone) of the upper ocean focusing on the 11 North American Large Marine Ecosystems (LMEs; Fig. 1). A decadal prediction system with embedded ocean biogeochemistry entitled the Community Earth System Model - Decadal Prediction Large Ensemble (CESM-DPLE)[22] is employed, which includes a Forced Ocean-Sea Ice (FOSI) reconstruction for prediction initialization ("Methods" section). The reconstruction is skillful in representing observed spatial and temporal variability of $T$ and $O_2$ in the upper-200 m ocean, although biases exist (Supplementary Fig. 2). Given the temporally and spatially sparse $O_2$ observations, we evaluate the initialized CESM-DPLE forecasts against the FOSI reconstruction instead of real observations, so the predictability in this work actually stands for potential predictability, an upper limit of predictability we could actually obtain[21,23,24]. This decadal prediction system has shown significantly enhanced predictive capacity of physical,

biogeochemical, and ecological variables in the ocean[21,22,24]. Our results suggest that the $\phi$ is potentially predictable in the upper-600 m ocean, showing a substantial improvement in average predictability timescale from 2 to 6 years against a simple persistence forecast (Fig. 1). Except over some high-latitude coastal regions, predictability of the $\phi$ is mainly attributed to its $O_2$ component rather than the $T$ or salinity ($S$) component for the ecotype considered. Distinct differences in predictability exist among ecotypes with different $E_o$ traits in the LMEs, especially along the northeast coast with the average predictability timescales ranging from 2 to 10 years for low-$E_o$ species.

## Results

### Interannual viable habitat shifts indicated by $\phi$

We focus on viable habitats of marine species with different $E_o$ traits (across the range of −0.2 to 1.0 eV) and the same $A_c$ trait (medium value of 10 atm$^{-1}$), as the $A_c$ trait works as a linear scaling coefficient for $\phi$ (its impact on $\phi$ is homogeneous in temporal and spatial dimensions). We refer to each trait combination ($A_c$ and $E_o$) as an 'ecotype', each of which can characterize multiple named and recognized species. We first select three representative ecotypes with the $E_o$ traits equal to −0.2, 0.4, and 1.0 eV, termed low (e.g., sea squirt, a cosmopolitan tunicate), medium (e.g., common littoral crab), and high (e.g., northern/deep-water shrimp) $E_o$ ecotypes, respectively. These traits correspond to different viable habitat distributions in space, as well as their interannual shifts, integrated with spatial differences and interannual variabilities of the environmental variables (e.g., $T$ and $O_2$; Fig. 2a–f).

The low-$E_o$ ecotypes mainly inhabit ($\phi > 1$) the subtropical regions covering the LMEs of Insular Pacific Hawaiian (IPH), Gulf of Mexico, and Southeast U.S. Shelf (SEUS), instead of the southwest coast off Mexico (which has extremely low $O_2$) and northern high-latitude regions (which have generally high $O_2$ and low $T$), e.g., the LMEs of Eastern Bering Sea (EBS), Aleutian Islands, and Labrador-Newfoundland (LN; Fig. 2a). On the interannual timescale, the viable habitats of low-$E_o$ ecotypes have large meridional contractions or expansions in the northern high-latitudes and longitudinal shifts along the southwest coast off the North America (Fig. 2b).

In contrast, the medium- and high-$E_o$ ecotypes mainly inhabit the northern high-latitude regions as well as the North Pacific Current region (around 45°N; Fig. 2c–f), as they usually require lower critical $p_{O_2}$ to meet metabolic demand in these relatively colder regions, a pattern of which generally reflects low $T$ and abundant $O_2$. Southern regions that have either low $O_2$ or high $T$, have limited habitability for these ecotypes, whereas the low-$E_o$ ecotype has relatively more viable habitats. Thus, the interannual shifts of viable habitats for medium- and high-$E_o$ ecotypes are generally meridional, suggesting the ecotypes would have northward contractions of viable habitats with the southern boundaries retreated northward in response to warming or deoxygenation and opposite southward expansions of viable habitats in response to cooling or oxygenation.

Within all these LMEs, aerobic habitat viability varies with depth and ecotypes. Shoaling or deepening habitats occurs on the interannual timescale, in response to the interannual variations of environmental $O_2$ and $T$ (Fig. 2g, h). For the northwest coast LMEs, habitat viability of these ecotypes generally decrease as depth increases in the upper-600 m ocean, with the viable habitats extending from surface to deeper depths for higher-$E_o$ ecotypes. The interannual shoaling or deepening of viable habitats is due to the vertical lower boundaries, with higher-$E_o$ ecotypes having larger vertical shifts. The southwest coast LMEs show similar vertical profiles of $\phi$ as the northwest coast LMEs, but with shallower lower boundaries and less difference in interannual viable habitat shifts across different $E_o$ ecotypes. The southeast coast LMEs and the IPH LME suggest distinct layer preference for different $E_o$ ecotypes, with the upper-200 m layer more habitable for lower-$E_o$ ecotypes and the lower 200-600 m layer for

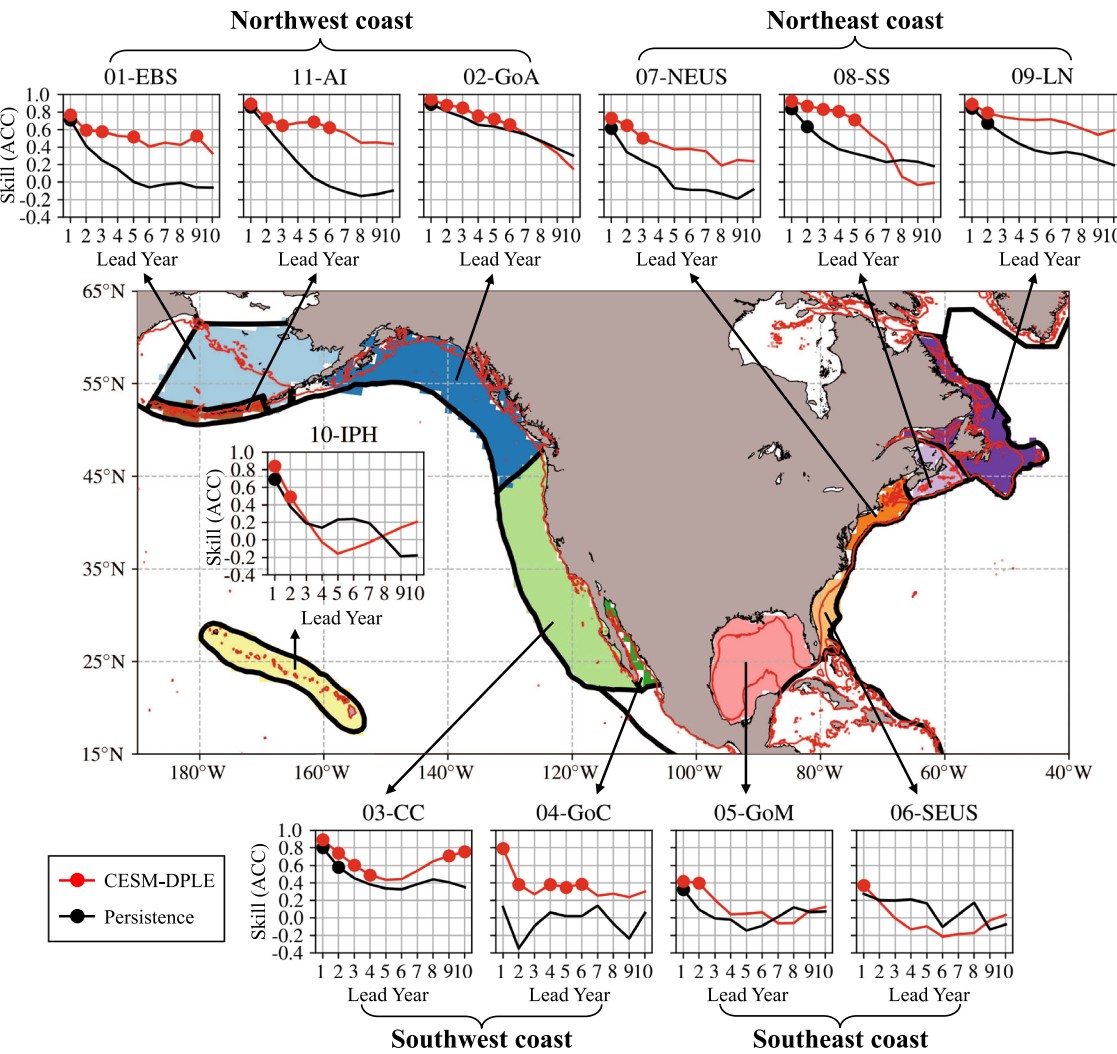

**Fig. 1 | Study domain and the eleven North American Large Marine Ecosystems (LMEs), with significantly higher prediction skills from the decadal prediction system against reconstruction persistence in predicting habitat viability in the upper-600 m ocean.** The prediction skills are assessed as Anomaly Correlation Coefficient (ACC). The decadal prediction system refers to the Community Earth System Model - Decadal Prediction Large Ensemble (CESM-DPLE)[22] system. The eleven LMEs are: 01-Eastern Bering Sea (EBS; 261 grid points), 02-Gulf of Alaska (GoA; 316 grid points), 03-California Current (CC; 378 grid points), 04-Gulf of California (GoC; 26 grid points), 05-Gulf of Mexico (GoM; 270 grid points), 06-Southeast U.S. Shelf (SEUS; 61 grid points), 07-Northeast U.S. Shelf (NEUS; 62 grid points), 08-Scotian Shelf (SS; 68 grid points), 09-Labrador-Newfoundland (LN; 263 grid points), 10-Insular Pacific Hawaiian (IPH; 163 grid points), and 11-Aleutian Islands (AI; 35 grid points). They are grouped into the northwest, northeast, southwest, and southeast coast LMEs, based on their geographical locations. The red contours on the map represent the 200-m isobath, based on bathymetry data from ETOPO1[54]. The boundaries of the LMEs are plotted as thick black lines on the map. For the panel of each LME, *y*-axis represents the prediction skill in ACC, and *x*-axis represents the lead year (LY) ranging from 1 to 10. The ACCs significantly nonzero at the 95% confidence level are marked with solid dots. Only ACCs of the medium-$E_o$ trait (temperature sensitivity of hypoxia vulnerability; 0.4 eV) species are presented. Data to reproduce the figures are shared on Figshare[53].

higher-$E_o$ ecotypes. Thus, viable habitats in these LMEs would generally shoal for lower-$E_o$ ecotypes, deepen for higher-$E_o$ ecotypes, and even with both upper and lower boundaries shrink inward for some medium-$E_o$ ecotypes (e.g., in the IPH LME), in response to warming or deoxygenation. Although lower-$E_o$ ecotypes have less temperature sensitivity, their viable habitats still vary strongly with $p_{O_2}$ on the interannual timescale. The northeast coast LMEs are habitable for almost all the $E_o$ ecotypes considered in the upper 600-m ocean, except those with low $E_o$ traits in the subsurface (below -50-m depth). However, their interannual viable habitat shifts can span hundreds of meters at depth, due to nearly vertical profiles of $\phi$ which are around the critical value of habitat viability ($\phi_{crit} = 1$).

### Interannual-to-decadal predictability of $\phi$

To assess the interannual-to-decadal predictability of habitat viability indicated by $\phi$, we present two metrics of prediction skill - the Anomaly

Correlation Coefficient (ACC; a measure of degree of associations and the most commonly used metric for forecast accuracy) and Normalized Mean Absolute Error (NMAE; a scale-normalized measure of distance or difference corroborating the ACC), between yearly anomalies of the CESM-DPLE ensemble-mean forecast and the FOSI reconstruction (formally, the predictand; "Methods" section). The resulting skills are then compared against those of a simple persistence forecast using the FOSI reconstruction, which measures the lowest predictability we can obtain ("Methods" section).

We first evaluate the interannual-to-decadal predictability of habitat viability for the medium-$E_o$ ecotype. The retrospective forecasts of $\phi$ have significantly (at the 95% confidence level; "Methods" section) higher predictability than the simple persistence forecasts in the upper-600 m ocean (Fig. 1), with improvement varying between regions, depths, and lead years (Figs. 3–4 and Supplementary Figs. 3–5). The two prediction skills - ACC and NMAE, suggest

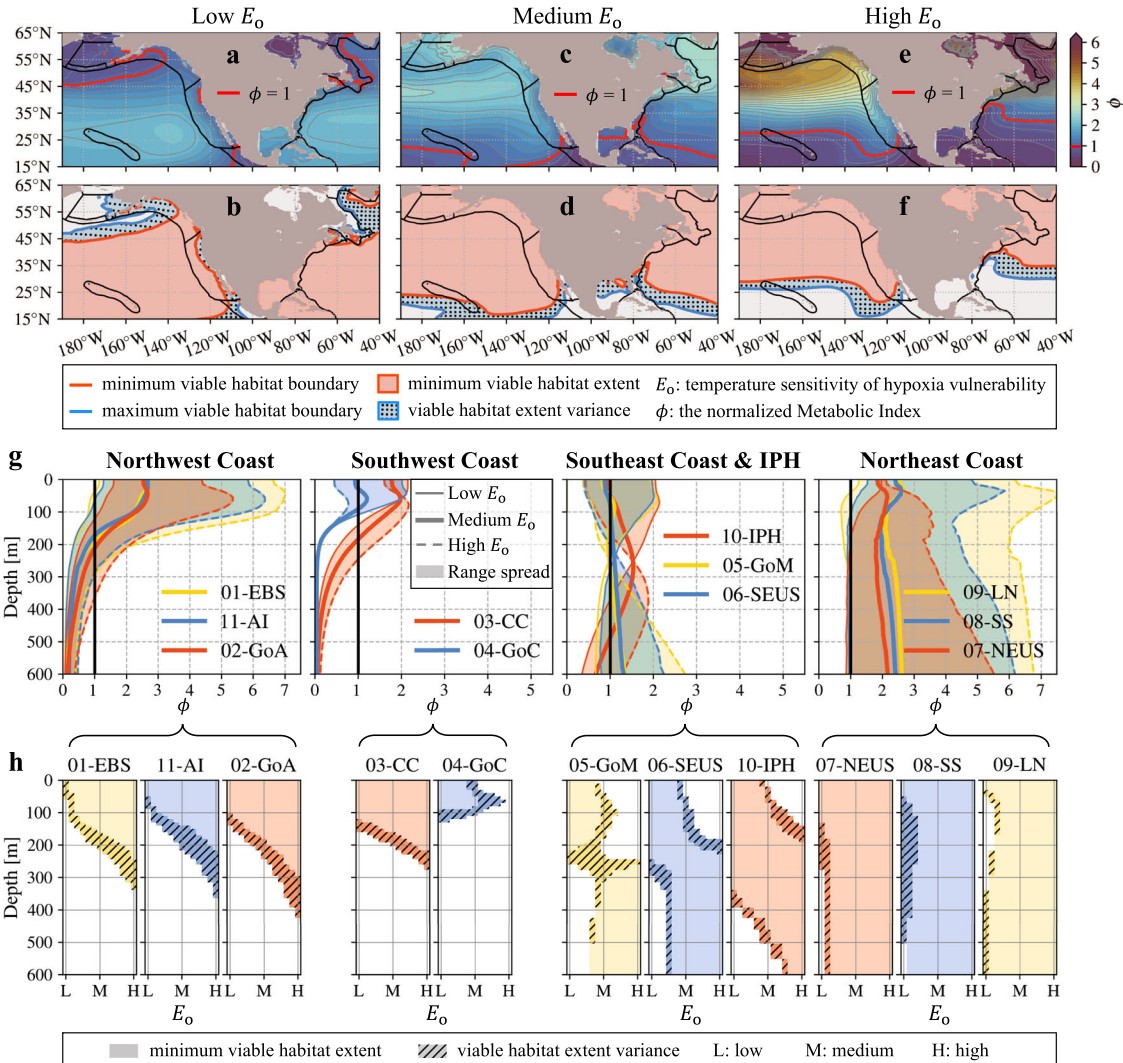

**Fig. 2 | Interannual viable habitat contraction or expansion in space and shoaling or deepening within the eleven Large Marine Ecosystems (LMEs).**
**a** Spatial pattern of the time-mean (1954-2017) normalized Metabolic Index ($\phi$) vertically averaged within the upper-200 m layer for the low $E_o$ (temperature sensitivity of hypoxia vulnerability) species ($E_o = -0.2$ eV, $A_c = 10$ atm$^{-1}$). The critical threshold value of $\phi$ ($\phi_{crit} = 1$) is plotted as red contours, setting the boundaries for habitable ($\phi > 1$) and uninhabitable ($\phi < 1$) regions in space. **b** Interannual shifts of viable habitats in space, with $\phi$ varying between $\phi \pm 3\sigma_\phi$, where $\sigma_\phi$ is the interannual standard deviation of $\phi$ over the period of 1954-2017. The red and blue contours represent the minimum and maximum viable habitat extents respectively, with the blue dotted shadows representing spatial difference in viable habitats on the interannual timescale. The minimum habitable regions are masked with red shadows. **c–f** Same as **a–b** but for the medium (0.4 eV; **c**, **d**) and high (1.0 eV; **e**, **f**) $E_o$ species (both $A_c = 10$ atm$^{-1}$). **g** Vertical profiles of the time-mean $\phi$ horizontally averaged within the LMEs (separated into four groups) for species with the $E_o$ traits ranging from low ($-0.2$ eV) to high (1.0 eV) values at a 0.1 eV interval (shadows). Each LME is represented by a different color as shown in the legend of each panel. The low, medium, and high $E_o$ species are plotted as thin solid, thick solid, and thin dashed lines for each LME, respectively. The x-axis represents the $\phi$ values and the vertical black line indicate the critical threshold value of $\phi$ ($\phi_{crit} = 1$). **h** Interannual contractions or expansions of viable habitats in the vertical direction within each LME for species with the $E_o$ traits ranging from low ($-0.2$ eV) to high values (1.0 eV) at a 0.1 eV interval (x-axis of each subpanel). Habitable depths are masked with shadows, and the hatched shadows represent depth ranges having habitat contractions or expansions on the interannual timescale. Each LME is represented by the same color as shown in the corresponding panels of **g**. Data to reproduce the figures are shared on Figshare[53].

consistent spatial and vertical patterns for the persistence and DPLE forecasts, with higher ACCs corresponding to lower NMAEs (Fig. 3). The persistence forecasts only have predictability timescale up to ~2 years in both depth layers of the eleven LMEs, with only a few spots that have longer timescales of predictability (~3 years), e.g., the northern Gulf of Alaska and the Labrador shelf and slope (Fig. 3a, b and Supplementary Fig. 3a, b). The DPLE forecasts provide significantly higher (at the 95% confidence level) prediction skills in almost all the LMEs, depth habitats, and lead years, with the average ACC improvement ( ΔACC) ranging from 0.1 in the IPH and SEUS to 0.4 in the Aleutian Islands (Supplementary Fig. 4). The average predictability timescales also increase up to 1 or 2 years in the IPH and SEUS and up to

~6 years in the Aleutian Islands using the DPLE forecasts. Although spatial difference in predictability exists within some LMEs, e.g., the mid-shelf of EBS, northern Gulf of Alaska, southern and northern California Current LME, and Labrador shelf have higher ACCs and longer predictability timescales against the rest regions of the corresponding LME depth habitats (Fig. 3c, d and Supplementary Figs. 3c, d and 4).

Improvement in prediction skill stems mainly from subsurface habitat, consistent with the preponderance of low-frequency variability in the ocean interior. For example, the DPLE forecast has achieved predictability timescale up to 9 years in the Gulf of Alaska at depths of 300-500 m and up to 7 years in the Aleutian Islands at depths

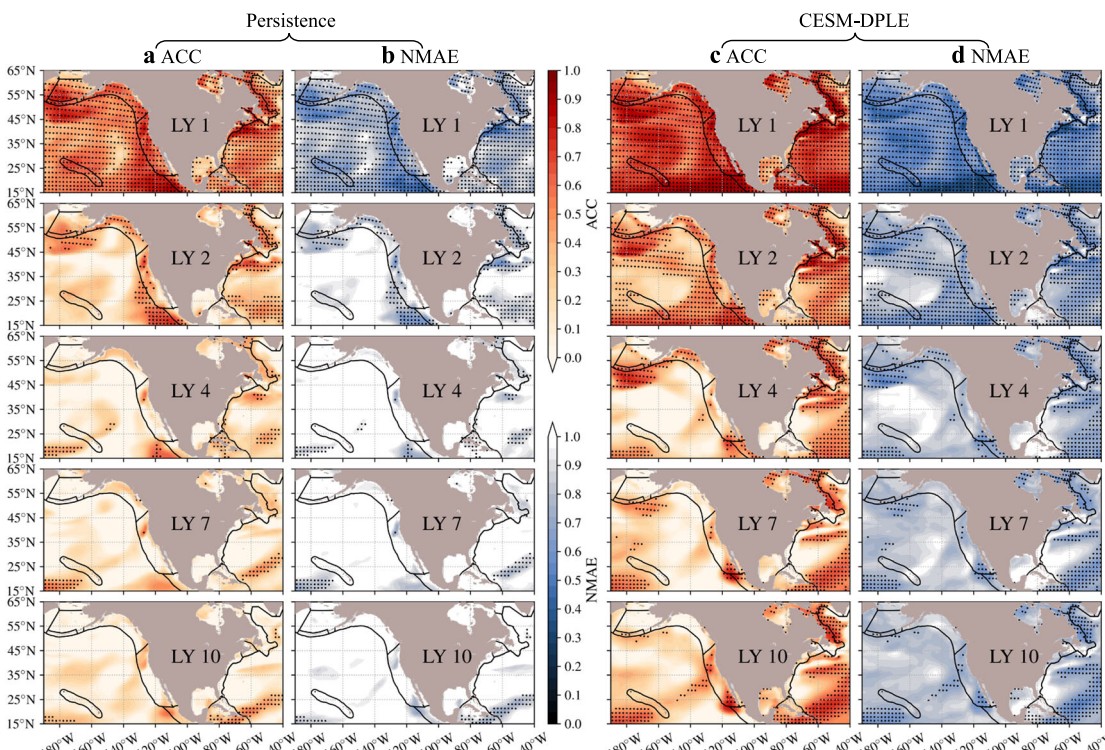

**Fig. 3 | Spatial patterns of significantly higher interannual-to-decadal prediction skills in habitat viability using the decadal prediction system at the upper-200 m layer for species with medium trait of temperature sensitivity of hypoxic vulnerability (0.4 eV). a, b** Prediction skills of Anomaly Correlation Coefficient (ACC; **a**) and Normalized Mean Absolute Error (NMAE; **b**) for the simple persistence forecast at lead year (LY) 1, 2, 4, 7, and 10. **c, d** Same as **a, b** but for the prediction skills of the Decadal Prediction Large Ensemble (DPLE) forecast. Black dots denote locations that have ACCs significantly nonzero at the 95% confidence level. The black solid contours in each panel indicate the boundaries of the Large Marine Ecosystems. Data to reproduce the figures are shared on Figshare[53].

of 100-300 m (Fig. 4). In the neighboring EBS LME, the DPLE forecast also has achieved higher prediction skills at depth than the persistence forecast, with the average predictability timescale of ~3 years against 1 year. The northeast coast LMEs not only have improved predictability from the DPLE at depth (below 100-m depth; up to ~4 years of the predictability timescale), but also higher prediction skills and longer timescales of predictability near the surface, e.g., up to ~7 years of predictability timescale in the upper-100 m of the Northeast U.S. Shelf (NEUS), Scotian Shelf, and the LN (mainly the Labrador shelf). The southeast coast LMEs and the IPH LME have substantial layer difference in prediction skills of the DPLE, with low prediction skills in ACC (~0.1) in the upper-200 m layer and much higher ACCs (~0.8) in the deeper 200-600 m layer. The significantly improved prediction skills (e.g., ACC increase by ~0.3, NMAE decrease by ~0.4, up to ~6 years against 1 or 2 years of predictability timescale in the SEUS) are mainly below the 200-m depth (Fig. 4 and Supplementary Fig. 5).

The LMEs of Gulf of California and California Current along the southwest coast suggest significant prediction skill improvement at depth as well (average ACC increase by ~0.4 and ~0.2, respectively), but the California Current LME exhibits a distinct prediction skill pattern from the DPLE forecast. Instead of steadily decreasing with longer lead years, prediction skill initially declines but rebounds at longer (4-10) lead years (Fig. 4e). The reemergent ACCs are located at ~100 m depth at lead year 4 and gradually extend deeper to ~300 m as the lead year increases to 10. The near-surface (<50 m) layer of the IPH LME also suggest similar (but weaker) reemergent ACCs at longer (4-10) lead years. These two LMEs' reemergence in prediction skills at 4 to 10 lead years is probably associated with the mean gyre-scale circulation in the North Pacific, i.e., mean advection of subsurface water mass anomalies[25]. Outside of these LMEs, the Labrador Sea (up to 6 years) and some open ocean regions (e.g., up to 9 years in the region south of Aleutian Islands) also exhibit long predictability timescales of the $\phi$,

indicating the advantage of using the DPLE forecasts in these regions as well.

## Mechanistic controls on interannual-to-decadal predictability

We investigate which driver variable lends the predictability to $\phi$ for the medium-$E_o$ ecotype by linearly decomposing $\phi$ into all possible driver variable components - $O_2$, $T$, and $S$ (see "Taylor Linear Decomposition" in "Method" section) and assessing these components' prediction skills relative to those of the $\phi$. We find that the $O_2$ component is the major contributor to the predictability of $\phi$ in most regions, depths, and lead years (Fig. 5 and Supplementary Fig. 6-7). However, for some LMEs, predictability of the $O_2$ component is very limited compared to that of $\phi$, for example, the inner shelves of EBS and the southeast coast LMEs (Gulf of Mexico and SEUS) at shorter (1-2) lead years, and the Labrador shelf at all lead years (Fig. 5a, d and Supplementary Figs. 6a and 7a). Outside of the LMEs, limited predictability of the $O_2$ component is also found in the upper-200 m layer of the Atlantic subtropical coastal regions (e.g., the Caribbean Sea) and both depth layers of the Labrador Sea (Supplementary Figs. 6a, b and 7a, b). The contribution to the predictability of $\phi$ from its $T$ component cannot be neglected in these regions, especially in the northern high-latitude regions along the inner shelf of the EBS and the Labrador shelf where the $T$ component even becomes the dominant driver variable (Fig. 5b and Supplementary Figs. 6c and 7c). The $T$ component contributes comparative predictability to $\phi$ with the $O_2$ component in some other regions as well, though mainly within the near-surface (<100 m) layer, for example, the northeast (NEUS and Scotian Shelf) and southwest (California Current and Gulf of California) coast LMEs and the IPH LME (Fig. 5b, e). The impact from the $S$ component, which arises from impact of $S$ on $O_2$ solubility ($O_2^{sol}$) in calculation of $p_{O_2}$, is more

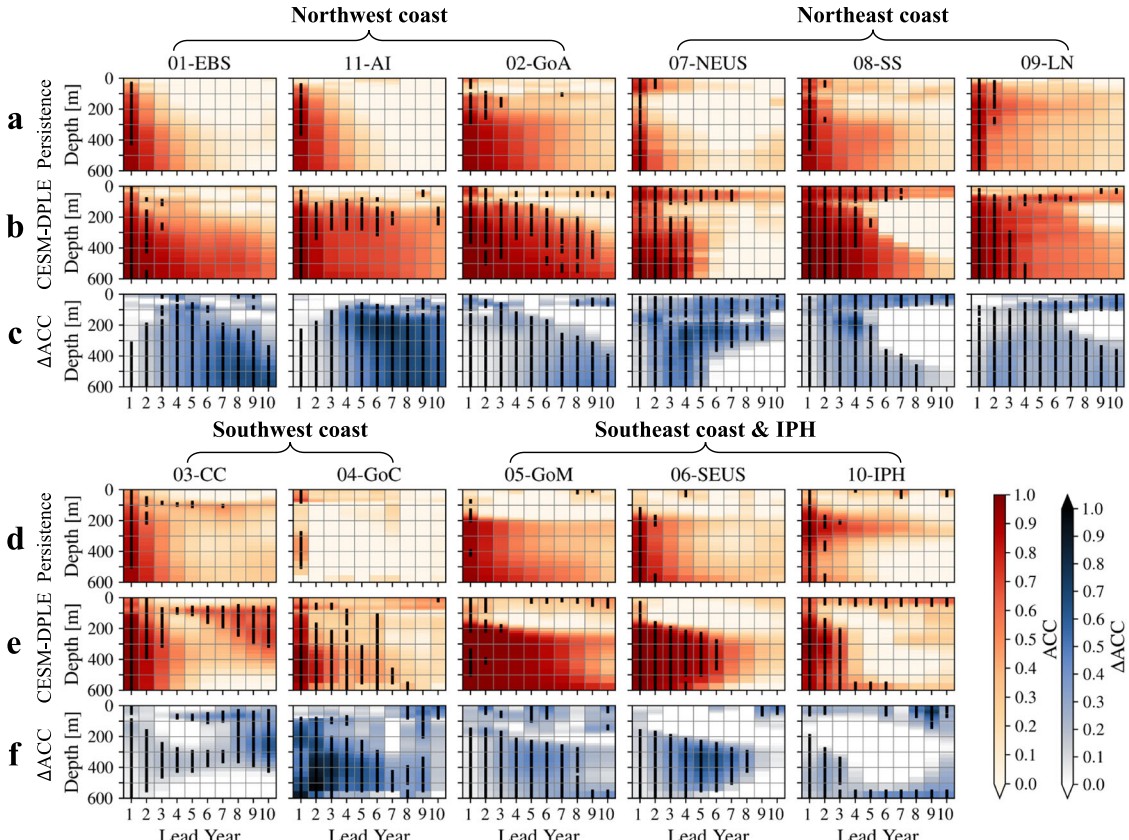

**Fig. 4 | Vertical distribution of significantly higher interannual-to-decadal prediction skills in habitat viability within the Large Marine Ecosystems (LMEs) using the Community Earth System Model-Decadal Prediction Large Ensemble (CESM-DPLE) forecast. a**, **b** Prediction skills of Anomaly Correlation Coefficient (ACC) for the medium-$E_o$ (temperature sensitivity of hypoxic vulnerability; 0.4 eV) species from the persistence (**a**) and DPLE (**b**) forecasts in the northwest (01-EBS, 11-AI, and 02-GoA) and northeast (07-NEUS, 08-SS, and 09-LN) coast Large Marine Ecosystems (LMEs). The x-axis of each panel represents lead year ranging from 1 to 10, and the y-axis represents depth from 0 to 600 m. At each lead year, depths that have significantly nonzero ACCs (at the 95% confidence level) are marked with thick black vertical lines. **c** Same as **a** or **b**, but for the ACC difference (ΔACC) between the DPLE and persistence forecasts (DPLE minus persistence). The significantly increased ACCs (at the 95% confidence level) are marked with thick black vertical lines at each lead year and in each panel. **d**–**f** Same as **a**–**c**, but for the southwest (03-CC and 04-GoC) and southeast coast (05-GoM and 06-SEUS) LMEs and the Insular Pacific Hawaiian (10-IPH). Data to reproduce the figures are shared on Figshare[53].

limited compared to those of $O_2$ and $T$ and mainly confined to the Labrador shelf and sea (Fig. 5c, f and Supplementary Figs. 6d and 7d).

A variance budget analysis of $\phi$ and its components ("Methods" section) further confirms the dominant role of the $O_2$ component in most regions of the study domain and the nonnegligible impacts from the $T$ component along the inner shelf of EBS and the northeast coast (Scotian Shelf and LN), where interannual variance of the $O_2$ and $T$ components is partly contradicted by their negative covariance at both depth habitats (Fig. 6a–e and Supplementary Fig. 8a–e). A further decomposition of the $O_2$ component into contributions associated with $O_2^{sol}$ and apparent oxygen utilization (AOU, "Methods" section) suggests that the AOU is the dominant component in driving the interannual variability of the $O_2$ component at both depth habitats in the Pacific regions (Fig. 6f–h and Supplementary Fig. 8f–h). While for the northeast coast regions in the Atlantic, interannual variability of the $O_2$ component is jointly contributed by the $O_2^{sol}$ and AOU components as well as their covariance at both depth habitats. Variance of the $O_2^{sol}$ component has relatively larger magnitude than that of the AOU component at the upper-200 m layer of the northeast coast regions, which is similar for the further decomposition of the covariance between the $O_2$ and $T$ components at these two layers (Fig. 6i–j and Supplementary Fig. 8f–j). Thus, the $O_2$ variability, especially the AOU variability, are critical factors for skillful predictions of $\phi$ for the medium-$E_o$ ecotypes in most regions on the interannual-to-decadal timescale; while in those high-latitude coastal regions - the EBS inner

shelf and the northeast coast regions (especially Scotian Shelf and LN), the physical drivers (variability of $T$ and $T$- and $S$- dependent $O_2^{sol}$) can be equally important and cannot be neglected, especially in the near-surface (<100 m) layer.

## Predictability differences across different $E_o$ ecotypes

The interannual-to-decadal predictability of aerobic habitat viability may vary largely among different marine ecotypes, due to their difference in the temperature sensitivity of hypoxia tolerance (i.e., $E_o$). We calculate prediction skill for ecotypes across a broad range of the $E_o$ trait (from −0.2 to 1.0 eV with an interval of 0.1 eV) and find that the predictability differences among different $E_o$ ecotypes are most notable in the northeast coast LMEs (Fig. 7 and Supplementary Fig. 9). For example, the upper depth layers of the northeast coast LMEs have predictability timescale up to only 1 or 2 years for ecotypes with the $E_o$ traits equal to 0.1-0.3, 0.2-0.3, and 0.6-1.0 eV for the NEUS, Scotian Shelf, and LN, respectively. But they have much longer predictability timescales for some higher- or lower- $E_o$ ecotypes, e.g., up to 10 years in the NEUS LME for ecotypes with the $E_o$ equal to 0 eV (Fig. 7a). The deeper depth layer of the northeast coast LMEs maintains the predictability differences similar to those of the upper layer. For example, the deeper layer of the NEUS has up to 4 years of predictability timescale for medium-to-high $E_o$ (above 0.2 eV) ecotypes but has up to 8-10 years for low-$E_o$ (below 0.1 eV) ecotypes (Supplementary Fig. 9a).

The upper depth layer of the northwest coast LMEs and the deeper depth layer of the SEUS LME also have notable predictability

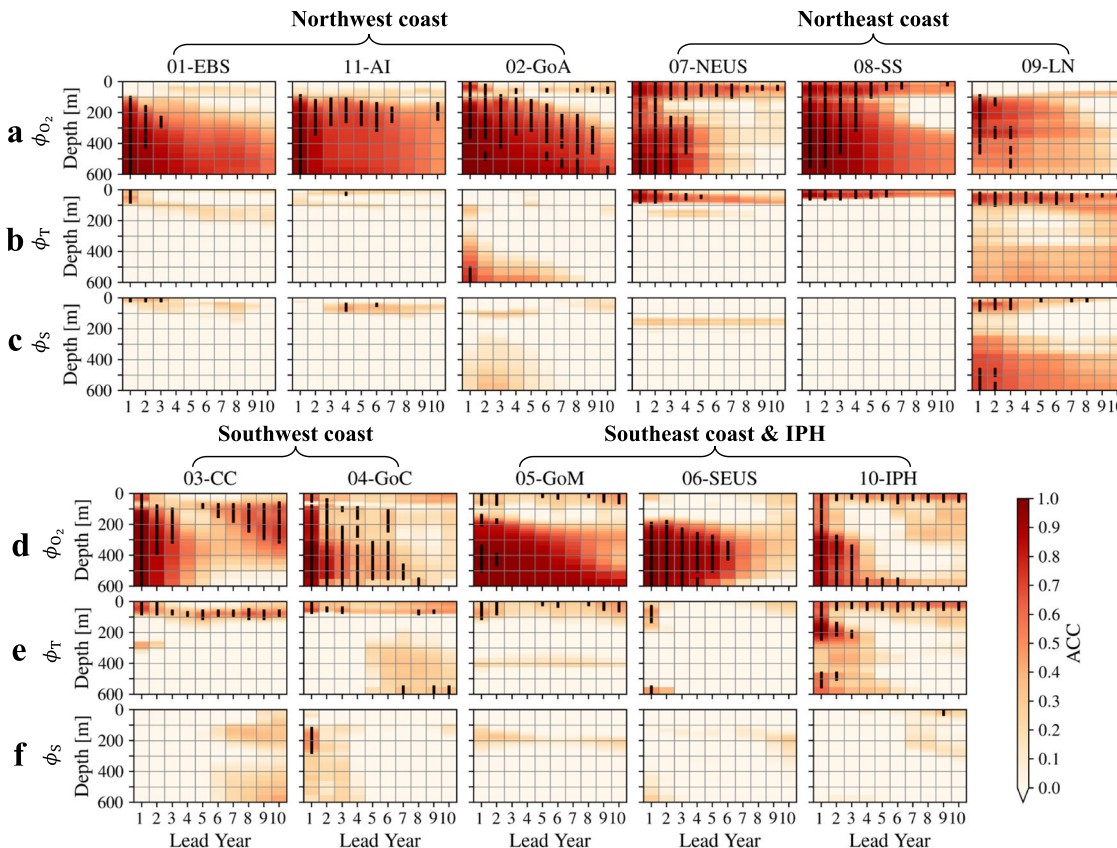

**Fig. 5 | Interannual-to-decadal predictability of habitat viability assessed as the Anomaly Correlation Coefficient (ACC) is dominantly contributed by the oxygen components in most ecosystems. a** Predictability of the linearly decomposed oxygen component ($\phi_{O_2}$) of the normalized Metabolic Index ($\phi$) using the DPLE forecasts for the medium-$E_o$ (temperature sensitivity of hypoxic vulnerability; 0.4 eV) species in the northwest and northeast coast Large Marine Ecosystems (LMEs). In each panel, depths that have statistically significant ACCs (nonzero at the 95% confidence level) are marked with thick black vertical lines. **b, c** Same as panel **a** but for the linearly decomposed temperature ($\phi_T$) and salinity ($\phi_S$) components of $\phi$. **d–f** Same as panels **a–c** but for the southwest and southeast coast LMEs and the Insular Pacific Hawaiian (IPH). Data to reproduce the figures are shared on Figshare[53].

differences across different $E_o$ ecotypes. The former has longer predictability timescale up to ~6 years for medium-$E_o$ ecotypes (0.4, 0-0.4, and 0.1-0.2 eV for the LMEs of EBS, Aleutian Islands and Gulf of Alaska, respectively) and very limited predictability timescale up to only 1-2 years for low- and high-$E_o$ ecotypes (e.g., $E_o$ below 0 and −0.1 eV in the LMEs of EBS and Gulf of Alaska, respectively, and above 0.5, 0.7, 0.8 eV in the LMEs of EBS, Aleutian Islands and Gulf of Alaska, respectively; Fig. 7a). Similarly, the latter has longer predictability timescale up to 6-7 years for medium-$E_o$ ecotypes (0.1–0.6 eV) and limited predictability timescale up to only 3-4 years for low- and high-$E_o$ ecotypes (below −0.1 eV and above 0.8 eV; Fig. 7e and Supplementary Fig. 9e). The rest depth habitats of LMEs have relatively little differences (less than 1 or 2 years) in predictability (especially prediction skills) across different $E_o$ ecotypes, although significance tests of prediction skills may result in some difference in the predictability timescales (e.g., in the southwest coast LMEs).

The mechanistic controls to the predictability differences across different $E_o$ ecotypes in these regions can be explained by relative contributions of the driver variable ($O_2$, $T$, and $S$) components. Generally, the $O_2$ component is the major contributor to the predictability of $\phi$ in most LMEs (especially for the lower depth layer of the Pacific LMEs), but not for all the ecotypes across this $E_o$ range (Fig. 7 and Supplementary Fig. 9). For example, in the upper-200 m of the northwest coast LMEs, the trait-space predictability differences are mainly contributed by the $O_2$ component, especially for the medium-$E_o$ ecotypes. The $T$ component also contributes to the predictability but only for the low- and high-$E_o$ ecotypes (e.g., below 0.1 eV and

above 0.8 eV in the LME of EBS with limited predictability timescale of 1 or 2 years; Fig. 7a–d). Similarly, at the deeper depth layer of the SEUS, the higher prediction skills (up to 6 years of predictability timescale) for medium-$E_o$ ecotypes (0.2-0.4 eV) are mainly contributed by the $O_2$ component. The $T$ component does contribute to the predictability but only for relatively low- (below −0.1 eV) and high- (above 0.8 eV) $E_o$ ecotypes at short-time (1 or 2) lead years (Supplementary Fig. 9e–h).

In contrast to other regions, in the northeast coast LMEs, the $T$ component plays a strong role in trait-space predictability differences (Fig. 7a–d and Supplementary Fig. 9a–d). The $O_2$ component only has comparative predictability (to that of the $\phi$) for habitat predictions of some medium-to-high $E_o$ ecotypes, e.g., above 0.2, 0.4, and 0.6 eV in the upper layer of NEUS, Scotian Shelf, and LN, respectively. The $S$ component also presents as an important contributor to the predictability in these LMEs, but only for some relatively low-$E_o$ ecotypes, e.g., below 0.2, 0.3, and 0.4 eV in the upper-200 m of NEUS, Scotian Shelf, and LN, respectively. It suggests a substantial impact from the $T$- and $S$-dependent $O_2^{sol}$ on the predictability of habitat viability for these relatively low $E_o$ ecotypes. The dominance of the $O_2$ or $T$ component in predictability differences of some southwest and southeast coast LMEs is generally less clear than the northern high-latitude LMEs, due in part to the overall limited predictability of $\phi$ (e.g., the southeast coast LMEs and the IPH LME at the upper-200 m layer). The $T$ component, however, does contribute to the predictability in the LMEs of California Current and Gulf of California at the upper-200 m layer, especially for relatively high $E_o$ ecotypes (e.g., above 0.6 eV in the Gulf of California; Fig. 7e–h).

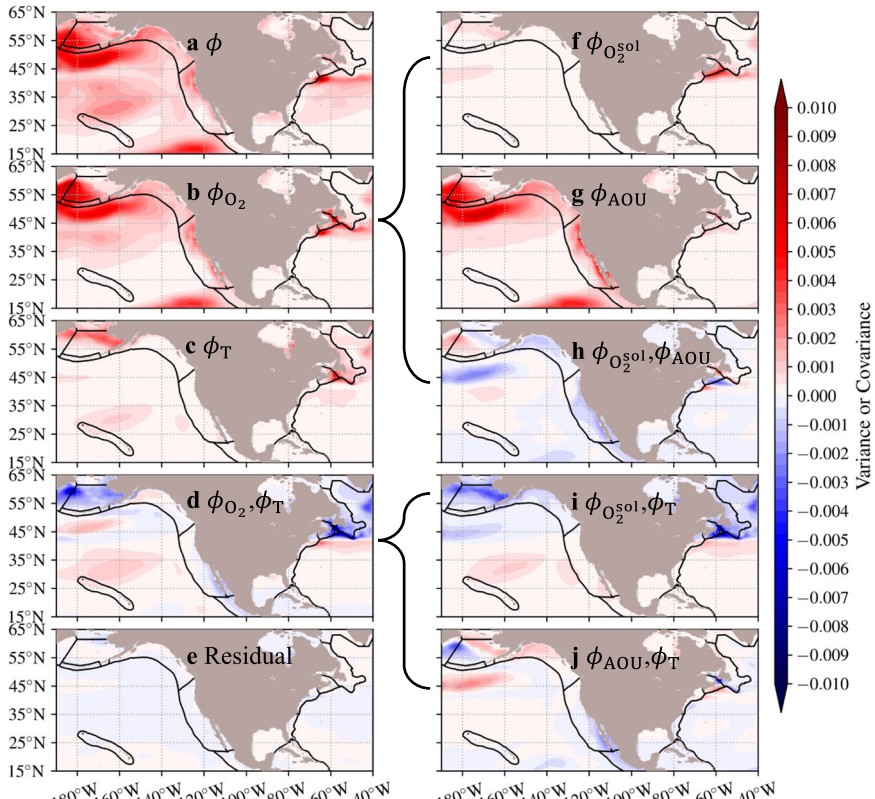

**Fig. 6 | Interannual variance of the normalized Metabolic Index ($\phi$) is mainly contributed by its oxygen-related components in the Pacific.**
**a**–**e** Decomposition of interannual variance of the normalized Metabolic Index ($\phi$; **a**) into contributions from the dissolved oxygen ($\phi_{O_2}$; **b**), temperature ($\phi_T$; **c**), and their covariance (between $\phi_{O_2}$ and $\phi_T$; **d**) components, as well as a residual term (**e**) at the upper-200 m depth habitat, based on Eq. (7). **f**–**h** The variance of the $\phi_{O_2}$ is

further decomposed into three components: the oxygen solubility ($\phi_{O_2^{sol}}$; **f**), apparent oxygen utilization ($\phi_{AOU}$; **g**), and their covariance between $\phi_{O_2^{sol}}$ and $\phi_{AOU}$ (**h**), based on Eq. (8). **i, j** The covariance term between $\phi_{O_2}$ and $\phi_T$ is also decomposed into two covariance terms between $\phi_{O_2^{sol}}$ and $\phi_T$ (**i**), and between $\phi_{AOU}$ and $\phi_T$ (**j**), based on Eq. (9). Data to reproduce the figures are shared on Figshare[53].

## Discussion

Recent advances in modeling and forecasting the earth system (e.g., refs.14,26,27) enable rapid expansion of marine ecological and habitat forecasts. Such forecasts have typically focused on $T$ as the primary if not sole determinant of ecological niche (e.g., ref. 28). This study expands the scope and skill of these tools by highlighting the potential that $O_2$ can play in deriving forecasts of marine habitats in the upper-600 m of the ocean. A key normalized Metabolic Index ($\phi$) enables quantifying habitat constraints for different species arising from their metabolic dependence on $T$ and requirements for $O_2$[3,10]. We combine this eco-physiologically mechanistic framework with a full suite of initialized retrospective decadal prediction system embedded with ocean biogeochemistry, to investigate the interannual-to-decadal predictability of aerobic habitat viability ($\phi$) for diverse marine ecotypes, its mechanistic linkages with the environmental driver variables, and the predictability differences across different $E_o$ ecotypes.

We select three representative marine ecotypes with low (−0.2 eV; e.g., sea squirt), medium (0.4 eV; e.g., common littoral crab), and high (1.0 eV; e.g., northern/deep-water shrimp) $E_o$ traits to demonstrate the model performance in capturing the interannual-to-decadal habitat shifts, which exhibit large spatial differences in the habitat viability ($\phi$) among different ecotypes, as well as their habitat shifts on this timescale. Due to spatial differences in the environmental driver variables (e.g., $O_2$, $T$, and $S$), the high-latitude coastal regions have high habitat viability with deep vertical extension for the medium- and high-$E_o$ ecotypes but limited viability for the low-$E_o$ ecotypes (Fig. 2), while the subtropical regions have the opposite (e.g., the Gulf of California) or layer-specified (e.g., the Gulf of Mexico and SEUS) habitat viability for

these ecotypes. In response to interannual variations of environmental variables, the viable habitats of medium- and high-$E_o$ ecotypes mainly have their southern boundaries shifted meridionally (e.g., retreat northward in response to warming or deoxygenation), while those of the low-$E_o$ ecotypes have both northern and southern or southeastern boundaries shifted not only meridionally but zonally (e.g., along the west coast of California). The habitat contraction or expansion trends are consistent with the observed relocation of various economically important species along both west and east coasts of North America. For example, the medium-$E_o$ species of American lobster and summer flounder are both shifting northward along the NEUS during recent years (e.g., refs. 29–31).

Combined with this mechanistic framework, the decadal prediction system is used to evaluate the interannual-to-decadal predictability of habitat viability at both depth habitats and vertically within the LMEs, providing significantly higher (at the 95% confidence level) prediction skills and longer predictability timescales than the simple persistence forecast. For marine ecotypes with the medium-$E_o$ trait, prediction skills of the DPLE are spatially inconsistent, with higher ACCs over the mid-shelf of EBS, northern Gulf of Alaska, southern and northern California Current LME, and the Labrador shelf (Fig. 3). Vertically, these improved prediction skills are mainly in the subsurface, e.g., below 100-m depth for the west (northwest and southwest) coast LMEs and 200-m depth for the southeast coast LMEs (Fig. 4). For habitat viability in the northeast coast LMEs, the DPLE has achieved both higher ACCs at depth and in the near-surface (<100 m) layer. For the LMEs of California Current and IPH, the decadal prediction system not only has achieved higher prediction skills at shorter (1-3) lead years

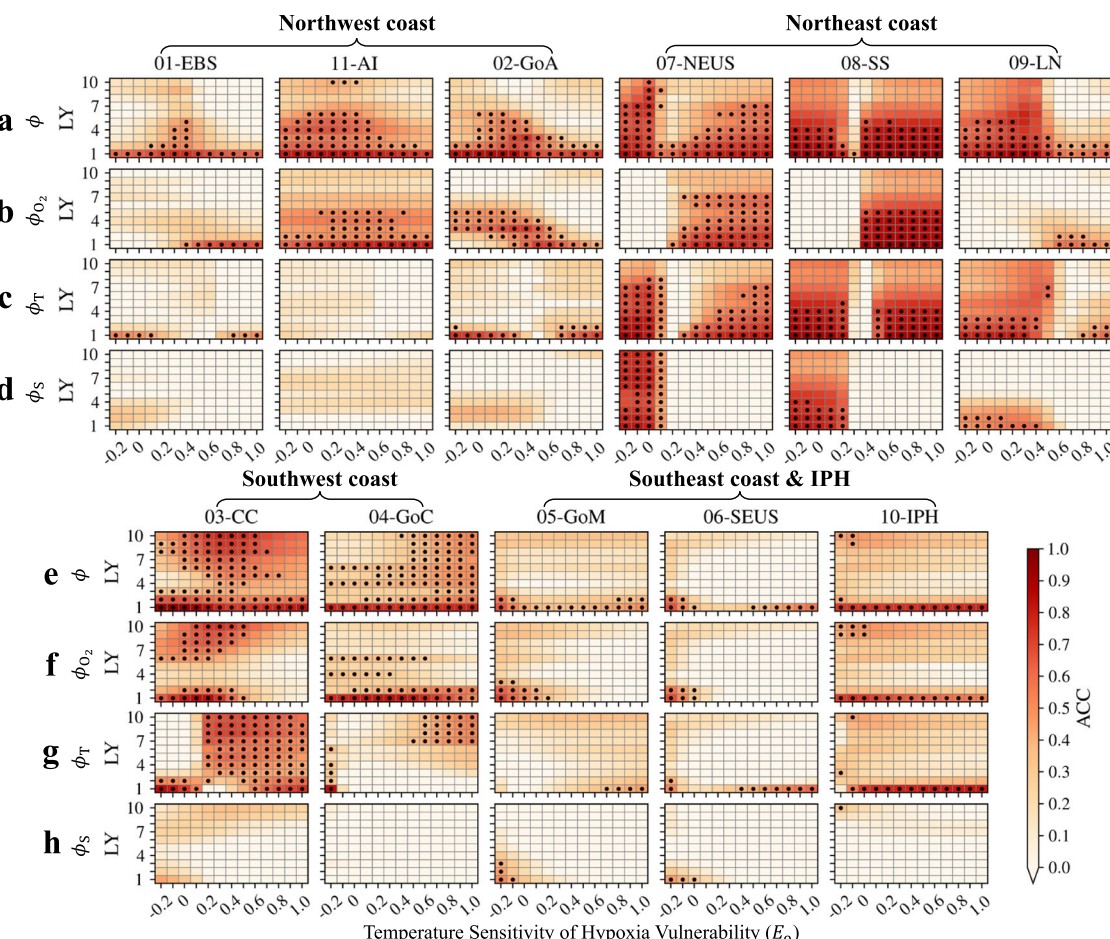

**Fig. 7 | Differences in predictability of habitat viability ($\phi$) among different $E_o$ (temperature sensitivity of hypoxia vulnerability) species and those of the oxygen ($\phi_{O_2}$), temperature ($\phi_T$), and salinity ($\phi_S$) components, in the upper-200 m of the eleven Large Marine Ecosystems (LMEs) using the decadal prediction system. a** Difference in predictability of the $\phi$ assessed as Anomaly Correlation Coefficient (ACC) among different $E_o$ species in the northwest and northeast coast LMEs. The $x$-axis of each panel represents the $E_o$ trait ranging from −0.2 to 1.0 eV at 0.1 eV interval, and the $y$-axis represents the lead year ranging from 1 to 10. The black dots in each panel suggest that the prediction skills in ACC are significant nonzero at the 95% confidence level. **b–d** Same as panel **a** but for the linearly decomposed oxygen ($\phi_{O_2}$), temperature ($\phi_T$), and salinity ($\phi_S$) components of the $\phi$. **e–h** Same as panels **a–d** but for the southwest and southeast coast LMEs and the Insular Pacific Hawaiian (IPH). Data to reproduce the figures are shared on Figshare[53].

but realized reemergent skillful ACCs at lead years from 4 to 10. These reemergent ACCs may be associated with the mean advection of subsurface water mass anomalies in the North Pacific at comparable timescales and warrants further dynamical investigation.

The decadal prediction system combined with the mechanistic framework also enables investigating the relative mechanistic controls to the predictability of habitat viability from the environmental driver variables ($O_2$, $T$, and $S$). The Taylor linear decomposition and variance budget analysis suggest that the $O_2$ component of $\phi$ is the primary control on the predictability in most regions (especially the Pacific) for the medium-$E_o$ ecotypes, which demonstrates the importance of reliable $O_2$ forecasts and measurements for skillful forecasts of habitat viability at a range of depths (e.g., ref. 32). However, contribution of the $T$ component to the predictability can also be important or even determinant over the high-latitude regions, e.g., the EBS inner shelf and the Labrador LME. This is probably because of relaxation of the $O_2$ constraint on habitat viability due to strong ventilation processes in these high-latitude regions. While the physical variables such as $T$ and $T$- and $S$- dependent $O_2^{sol}$ exert larger impacts on habitat viability in these regions, these physical impacts are mainly in the near-surface (<100 m) layer, i.e., in the northeast coast LMEs (Fig. 5). This work also identifies the importance of AOU for interannual-to-decadal habitat predictions. Improved simulation and representation of the processes

responsible for AOU variability - ventilation and organismal $O_2$ production and consumption, would also further improve our habitat forecasts. The former could be enhanced with higher resolutions, while the latter remains unresolved and requires more physiological, ecological, and modeling efforts.

This study also provides a trait-space view of predictability differences across marine ecotypes with different temperature sensitivities of hypoxia vulnerability - the $E_o$, which can be explained by the relative dominance of the environmental driver variables. The trait-space predictability differences are prominent in the upper-200 m of the northwest coast LMEs, both depth layers of the northeast coast LMEs, and lower 200-600 m of the southeast coast LMEs (Fig. 7 and Supplementary Fig. 9). It suggests predictability of habitat viability in these regions may need case-by-case investigation for certain species and the DPLE system may have extremely high performance in decadal habitat predictions of some species, e.g., low $E_o$ (0 eV) ecotypes in the NEUS LME. For most LMEs that have notable predictability differences (the upper layer of the northwest coast LMEs and the deeper layer of the SEUS LME), the $O_2$ component is the major contributor for medium-$E_o$ ecotypes, which suggests much longer predictability timescales than those of the $T$ component for low- and high-$E_o$ ecotypes. While predictability differences of the northeast coast LMEs are mainly contributed by the $T$ component (especially for low-to-medium $E_o$

ecotypes with up to 10 years of predictability timescale) and impacts from the $O_2$ component exist only for certain $E_o$ range (medium-to-high) of ecotypes. We focus on habitat shifts and viability predictions for different $E_o$ ecotypes with the same $A_c$ (10 atm$^{-1}$) trait. Due to the linear scaling effect of the $A_c$ trait, these results are applicable to any other ecotypes with the same $E_o$ but different $A_c$ traits. For example, the interannual habitat shifts are similar for ecotypes with the same $E_o$ but smaller (larger) $A_c$ traits, only having the habitat boundaries moving northward (southward) or towards high (low) $\phi$ regions. The prediction skill assessments in ACC are also the same between the same-$E_o$ different-$A_c$ ecotypes, as the linear scaling coefficient $A_c$ is self-canceled.

Our results demonstrate the potential of using an initialized ESM to retrospectively predict habitat viability for diverse marine ecotypes on the interannual-to-decadal timescale. With skillful interannual-to-decadal predictions of the $\phi$, spatial extents of viable habitats and the corresponding areas and volumes for diverse marine ecotypes can be estimated. The generated habitat forecast products can be fully employed for living marine resource management and decision making in response to the changing ocean conditions (e.g., ref. 33). For example, northward habitat contractions can make it infeasible to fish from certain ports, while southward habitat expansion opens ports that would need to be prepared for processing the landings. Changes in depth distribution can also necessitate changing the type of fishing gear used. Additionally, unequal spatial and vertical habitat shifts of species due to their different metabolic traits (e.g., temperature sensitivity) may lead to substantial changes in prey-predator dynamics and ecosystem structure (e.g., overlap of viable habitats of southern and northern silver hake with juveniles as prey and adults as predator, ref. 34), which may affect resource availability to the fishery and require adaption by stakeholders[33].

Although this study presents a promising result, it is important to note that the coarse spatial resolution (~1°) may not resolve well the fine-scale oceanic processes that are important for physical and biogeochemical variations. Higher-resolution global forecast models (e.g., the developing high-resolution CESM decadal forecasts at ~0.1°) or regional models dynamically downscaled from the global initialized ESM forecasts can be used to further improve the habitat viability forecasts, based on the framework of $\phi$. However, reliable observations or observation-based products of $O_2$ with spatio-temporal variability are urgently needed to be able to truly verify the realized prediction skills and move from potential predictability to verified prediction skill. These observational products exist for $T$, $S$, and chlorophyll, but are just beginning to be developed for $O_2$ (e.g., the GOBAI-O2 product, ref. 35) and other biogeochemical variables.

## Methods

### The Metabolic Index
The Metabolic Index ($\Phi$) is defined as the ratio of dissolved oxygen ($O_2$) supply to an organism's resting metabolic demand for evaluation of the aerobic energy balance of an organism, as expressed by this equation:

$$\Phi = \frac{\text{Supply}}{\text{Demand}} = A_o \frac{p_{O_2}}{\exp\left\{-\frac{E_o}{k_B}\left[\frac{1}{T} - \frac{1}{T_{ref}}\right]\right\}} \quad (1)$$

where $A_o$ is the ratio of efficacy for $O_2$ supply ($\alpha_S$) and minimum metabolic rate ($O_2$ demand; $\alpha_D$) at reference temperature ($T_{ref}$; $A_o = \frac{\alpha_S}{\alpha_D}$), $E_o$ is the difference between temperature variation in the metabolic rate ($E_d$) and that in the oxygen supply ($E_s$; $E_o = E_d$-$E_s$)[10], $p_{O_2}$ is the partial pressure of $O_2$ calculated using $O_2$ and temperature ($T$) and salinity ($S$) dependent oxygen solubility[36], and $k_B$ is the Boltzmann constant[37]. To account for the variation of $E_o$ with $T$, the $E_o$ value is modified by a linear term: $E_o \rightarrow E_o + \frac{dE_o}{dT}(T - T_{ref})$, with the $\frac{dE_o}{dT}$ equal to an average value of 0.022 eV/K for ecotypes considered in this study

and the $T_{ref}$ equal to 288.15 K[10]. Spatial distributions of this index compared with observed occurrence of various species with different $E_o$ traits can be found in the Supplementary Information of ref. 10.

We focus on the normalized Metabolic Index ($\phi$; $\phi = \Phi/\Phi_{crit}$), with the $A_o$ normalized to the $\Phi_{crit}$ at $T_{ref}$. If $\phi$ falls below 1, the environment can no longer support the aerobic demands of species' energetic requirements. Conversely, values of $\phi$ above 1 permit critical activities such as feeding, defense, growth, and reproduction.

To calculate depth and LME averages of the $\phi$, we first calculate the $\phi$ at each individual grid cell, and then perform depth-averaging at each layer and area-averaging within each LME, with depth- and area-weighted functions applied. To evaluate habitat viability (indicated by the $\phi$) on the interannual timescale, we first calculate the $\phi$ from monthly model outputs of $O_2$, $T$, and $S$, and then calculate yearly averages of the $\phi$ by averaging from January to December. To investigate interannual viable habitat shifts, we first calculate interannual standard deviation of the $\phi$ ($\sigma_\phi$) over the period of 1954–2017, and then identify the viable habitats ($\phi > 1$) in space and vertically within the LMEs, with the $\phi$ varying between $\phi \pm 3\sigma_\phi$. The corresponding spatial (vertical) differences in the viable habitats suggest habitat contraction or expansion (shoaling or deepening) on the interannual timescale.

### The Decadal Prediction System
The CESM-DPLE is a suite of initialized retrospective Earth System Model simulations with embedded ocean biogeochemistry[22]. Horizontal resolution of the CESM-DPLE ocean component is nominally $1° \times 1°$ with 60 vertical levels. The ocean and sea ice model components in the CESM-DPLE were re-initialized from a forced ocean-sea ice reconstruction (referred to as the FOSI reconstruction) each year on November 1st from 1954 to 2017[22]. For each initialization date, an ensemble of 40 forecast members was created by creating Gaussian perturbations to the initial atmospheric temperature field (order $10^{-14}$ K) at each grid cell integrating forward for ~10 years, and model components developed as a result of the spread in the atmospheric state[22].

The Biogeochemical Elemental Cycling (BEC) model is used to simulate biogeochemistry components in CESM-DPLE[38,39]. Note that the ocean biogeochemistry is diagnostic that has no feedback onto the simulated physical climate[22]. Historical radiative forcing with volcanic aerosols is included in CESM-DPLE through 2005, and projected radiative forcing (including greenhouse and short-lived gases and aerosols) is added from 2006 onward.

Drift adjustment is performed with CESM-DPLE, to correct for model drift caused by full-field initialization. The correction procedures are the same as described in ref. 22: the lead-time dependent model climatology is computed as the mean drift across all 40 ensemble members and 64 initialization dates between 1954 and 2017; and this drift was then subtracted at each grid cell from all forecasts to generate anomalies.

The FOSI reconstruction is a hindcast simulation from 1948 to 2017, with active ocean (physics and biogeochemistry) and sea ice model components from CESM that has identical grids as the fully coupled CESM-DPLE. It is forced at the surface by a modified version of the Coordinated Ocean-Ice Reference Experiment (CORE) with interannual forcing[40,41]. The FOSI reconstruction reproduces some key aspects of observed ocean (e.g., ocean temperature and dissolved oxygen as shown in Supplementary Fig. 2) and sea ice variability quite well, although there is no direct assimilation of either ocean or sea ice observations[42–44]. The FOSI reconstruction is used to evaluate the prediction skill of the simple persistence forecast, as a reference compared with the CESM-DPLE's prediction skill.

### Prediction skill assessments
Predictability is assessed by calculating the Anomaly Correlation Coefficient (ACC) and the Normalized Mean Absolute Error (NMAE)

between yearly anomalies from the CESM-DPLE ensemble-mean forecast and the FOSI reconstruction, both of which are considered as functions of lead time in years:

$$\text{ACC}(\tau) = \frac{\sum_{i=1}^{N} F'_i(\tau) \cdot R'_{i+\tau}}{\sqrt{\sum_{i=1}^{N}(F'_i(\tau))^2 \cdot \sum_{i=1}^{N}(R'_{i+\tau})^2}}, \quad (2)$$

$$\text{NMAE}(\tau) = \frac{1}{N}\sum_{i=1}^{N} \frac{|F'_i(\tau) - R'_{i+\tau}|}{\sigma_{R'}(\tau)}, \quad (3)$$

where $F'$ and $R'$ represent yearly anomalies of the DPLE ensemble-mean forecast and the FOSI reconstruction respectively, $\tau$ is the lead time in years ranging from 1 to 10, $N$ is the length of the time series, and $\sigma_{R'}$ is the interannual standard deviation of the $R'$ time series. The NMAE is considered as an accurate assessment of bias when quantifying the accuracy of the forecasts[21,45]. Statistical significance of ACC is tested nonzero at the 95% confidence level via a Student's $t$-test, considering the effective degree of freedom to account for autocorrelation of the two timeseries being correlated[46]. Higher values of ACC and lower values of NMAE suggest better prediction skills from the DPLE forecast.

To show the utility of the initialized forecasting system (DPLE) over a simple, low-cost forecasting method, we also evaluate these prediction skills of a persistence forecast using the FOSI reconstruction (hereafter named reconstruction persistence), which predicts the yearly anomalies of the FOSI reconstruction using predictors of itself but with $\tau$ lead years. The ACC and NMAE equations are the same as Eqs. (2) and (3), except replacing the $F'_i(\tau)$ with the $R'_i$. To assess if the prediction skill of the DPLE is better over that of the reconstruction persistence, we calculate statistical significance of the difference between these two ACCs (DPLE minus persistence), which is also tested larger than 0 at the 95% confidence level following ref. [47,48], taking into account the fact that the two correlations are based on a common FOSI reconstruction time series ($R'$)[49].

The predictability timescale is quantified as the maximum lead time when the corresponding ACCs are continuously significant (at the 95% confidence level) starting from lead year 1. For example, if ACC is statistically significant at lead year 1 through 5 and at lead year 8 through 9, the predictability timescale is 5 years instead of 9 years.

### Taylor linear decomposition and variance budget

To understand the mechanistic controls towards the predictability, a first-order Taylor-series decomposition method is applied to decompose the normalized Metabolic Index ($\phi$) into contributions from the physical ($T$, $S$) and biogeochemical ($O_2$) driver variables. For a certain marine species with known metabolic traits ($A_c$ and $E_o$), $\phi$ is a function of $O_2$, $T$, and $S$ (Eq. 1; $p_{O_2}$ is a function of $O_2$ and $T$- and $S$- dependent oxygen solubility), and its first-order Taylor-series decomposition is:

$$\phi = \bar{\phi} + \sum_{i=1}^{n} \frac{\partial \phi}{\partial x_i} \cdot (x_i - \bar{x}_i) + \text{residual} \quad (4)$$

where $x_i$ represents one of the three driver variables ($O_2$, $T$, and $S$; $n = 3$), $\bar{x}_i$ is the time-mean value of the variable $x_i$, and $\bar{\phi}$ is the mean value of $\phi$ calculated using all the time-mean driver variables. $\frac{\partial \phi}{\partial x_i}$ is the first-order partial derivative of $\phi$ with respect to the driver variable $x_i$. The residual term suggests higher-order terms that could be omitted. In other words, we decompose variations of $\phi$ into a constant time-mean term ($\bar{\phi}$) and three driver variable components:

$$\phi(t) \cong \bar{\phi} + \phi_{O_2}(t) + \phi_T(t) + \phi_S(t), \quad (5)$$

where $t$ represents time. The $\phi_{O_2}(t)$ could be further decomposed into an oxygen solubility component ($\phi_{O_2^{sol}}$) and an apparent oxygen utilization ($\phi_{AOU}$) component, depending on their linear relationship: $O_2$ is the difference between $O_2^{sol}$ and AOU (e.g., ref. 50; $O_2 = O_2^{sol} - \text{AOU}$, where $O_2^{sol}$ nonlinearly depends on $T$ and $S$, and AOU reflects the balance between respiration-driven oxygen consumption and ocean ventilation[32,36,51]):

$$\phi_{O_2}(t) = \phi_{O_2^{sol}}(t) + \phi_{AOU}(t), \quad (6)$$

where $\phi_{O_2^{sol}} = \frac{\partial \phi}{\partial O_2} \cdot (O_2^{sol} - \bar{O}_2^{sol})$, $\phi_{AOU} = -\frac{\partial \phi}{\partial O_2} \cdot (\text{AOU} - \overline{\text{AOU}})$, as $\frac{\partial \phi}{\partial O_2} = \frac{\partial \phi}{\partial O_2^{sol}} = -\frac{\partial \phi}{\partial \text{AOU}}$. We perform the same prediction skill assessments to evaluate the predictability of each physical and biogeochemical driver component at each depth habitat and LME. These driver components are calculated using the DPLE and considered as forecasts of the original, unchanged FOSI reconstruction. To better interpret the predictability of these driver components of $\phi$, we perform a variance budget analysis by decomposing the variance of $\phi$ ($\sigma_\phi^2$) into contributions from three driver components and their covariances, based on Eqs. (5) and (6):

$$\sigma_\phi^2 \cong \sum_{i=1}^{n} \sigma_{\phi_i}^2 + 2\sum_{j \neq i} \text{Cov}(\phi_j, \phi_i), \quad (7)$$

where $\phi_i$ is the driver component with $i$ representing one of the driver variables ($O_2$, $T$, and $S$; $n = 3$), and $\text{Cov}(\phi_j, \phi_i)$ represents the covariance between any two of the different driver components. The variance decomposition of $\phi_{O_2}$ also follows this rule, which is:

$$\sigma_{\phi_{O_2}}^2 = \sigma_{\phi_{O_2^{sol}}}^2 + \sigma_{\phi_{AOU}}^2 + 2\text{Cov}\left(\phi_{O_2^{sol}}, \phi_{AOU}\right) \quad (8)$$

Similarly, the covariance term between $O_2$ and $T$ components can also be decomposed as:

$$\text{Cov}(\phi_{O_2}, \phi_T) = \text{Cov}\left(\phi_{O_2^{sol}}, \phi_T\right) + \text{Cov}\left(\phi_{AOU}, \phi_T\right) \quad (9)$$

## Data availability
Output from the Community Earth System Model Decadal Prediction Large Ensemble (CESM-DPLE) and CESM model reconstruction can be downloaded at [https://www.earthsystemgrid.org/dataset/ucar.cgd.ccsm4.CESM1-CAM5-DP.html]. The metabolic traits data can be downloaded online at the supplementary table of ref. 10 [https://www.nature.com/articles/s41586-020-2721-y]. The GOBAI-O2 observation product[35] can be downloaded at [https://www.ncei.noaa.gov/access/metadata/landing-page/bin/iso?id=gov.noaa.nodc:0259304]. The EN4 temperature observation product[52] can be downloaded at [https://www.metoffice.gov.uk/hadobs/en4]. Source data of figures are provided with this paper and shared on Figshare[53].

## Code availability
The code used to calculate the normalized Metabolic Index, assess predictability, perform linear decomposition, and create all the figures are available on GitHub [https://github.com/zchenocean/viable-habitat-prediction].

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

## Acknowledgements

This work was supported by NOAA's Climate Program Office's Modeling, Analysis, Predictions, and Projections (MAPP) program NA20OAR4310437 (S.S.) and NOAA's funding project NA18NOS4780167 (C.D.). We acknowledge our participation in MAPP's Marine Ecosystem Task Force. This material is based upon work supported by the National Center for Atmospheric Research, which is a major facility sponsored by the National Science Foundation under Cooperative Agreement No. 1852977. The authors acknowledge helpful discussions and suggestions from Drs. J. Huang (WHOI) and A. C. Ross (GFDL). The authors also thank Dr. K. Krumhardt (NCAR) for her help on the CESM-DPLE forecasts. We also thank University of Connecticut Scholarship Facilitation Fund and Department of Marine Sciences for supporting the publication cost.

## Author contributions

Z.C., S.S., and M.L. assembled and analyzed the data. Z.C. wrote the manuscript. S.S. and M.L. supervised the study. Z.C., S.S., M.L., C.P., C.S., and C.D. interpreted the results and clarified the implications.

## Competing interests

The authors declare no competing interests.
