## [Peer Review File · Nature Communications]

Skillful Multiyear Prediction of Marine Habitat Shifts Jointly
Constrained by Ocean Temperature and Dissolved OxygenREVIEWER COMMENTS

Reviewer #1 (Remarks to the Author):

The manuscript "Skillful Multiyear Prediction of Marine Habitat Shifts Jointly Constrained by Ocean Temperature and Dissolved Oxygen" by Chen et al examines the potential predictability of species ecotypes using a physiologically-rooted approach.

This manuscript represents a major advance in our approach to ecological prediction in the ocean. Marine science has typically had very narrow focus on the environmental variables that shape the distribution of species, typically using temperature as the sole (or major) determinant of the ecological niche, and therefore to their prediction. This paper breaks with that tradition by highlighting the potential that oxygen can play in deriving predictions. It therefore represents a substantial breakthrough in the field. The paper is well written, convincing and without any major errors.

If, and only if, another revision of the manuscript is required, I would encourage a few minor clarifications that would improve the quality of the paper

- The author's use of the word "species" is a little too loose. The re-prediction exercise is not based on the distribution of an actual species, but on some "representative" parameter values that the authors have chosen. This is fine (and indeed eminently sensible!) but I would prefer that this point is made clearer. The author's are clearly aware of it, and suggest the use of the phrase "ecotype" on line 380 – this is a good alternative that could be more widely adopted throughout the manuscript, including the abstract.
- Similarly, it is important to note that this is an exercise in "potential predictability", as the "reference" dataset against which the predictions are compared is ultimately a product derived from the same model system. I understand why the author's have taken this approach – oxygen observations are rare – but this important detail is unfortunately buried in the methods. Again, I would like to see the authors be upfront about this, and adopt the language throughout the manuscript, including the abstract. A discussion of the impact of other oxygen products could be a useful addition to the discussion
- Finally, I am missing the link from the "ecotypes" back to the real species and their distribution, and ultimately to real-world predictions of species distributions. Given the decadal-time scales considered here, climate change is not really an important motivating factor – the work is more relevant to generating ecological forecast products for use in decision making. There is fortunately plenty of good work on this topic to cite e.g. Tommasi et al 2017, Payne et al 2017 and O'Kane et al 2023.

Long story short – this is a great article that should be published.

Mark R. Payne

Danish Meteorological Institute (DMI)

Copenhagen, Denmark

References:

Tommasi et al 2017 <https://www.sciencedirect.com/science/article/abs/pii/S0079661116301586>

Payne et al 2017 <https://www.frontiersin.org/articles/10.3389/fmars.2017.00289/full>

O'Kane et al 2023 doi: 10.3389/fclim.2023.1121626

Reviewer #2 (Remarks to the Author):

Co - reviewed with Reviewer #3

Reviewer #3 (Remarks to the Author):

Broad overview

We appreciated the opportunity to review a “Skillful Multiyear Prediction of Marine Habitat Shifts Jointly Constrained by Ocean Temperature and Dissolved Oxygen.” The manuscript validates a critical time scale of predictability from interannual to decadal scales – using a full suite of models to look beyond temperature to oxygen, and combining the two to address organismal requirements. The premise of the study is worthy of publication but the manuscript itself should be revised further before acceptance. Our biggest concerns focus on the precision of the writing. Specifically, we are not getting information on the results of differences that allow the reader to assess whether the findings are significant or not and instead have to trust on the author’s expertise. It’s a pervasive issue throughout the manuscript but an easy topic to address! For the methods, a table outlining the equations and variables that make them up would be more useful than explaining everything in the main body of the text and would reduce overall complexity and clutter. This would also provide a one stop location to quickly review terms while interpreting results. We suggest avoiding the use of acronyms as much as possible and instead spelling out names if something is only mentioned a couple times as there are many abbreviations to keep track of. We appreciated the section starting on L203 which digs into the mechanistic drivers of predictability as it helps the manuscript be of broad interest to the journal’s readership. Overall however, we had a hard time digging into the take-home messages because of the lack of specificity in writing but feel that if the writing were more clear, the manuscript could be a good and important contribution to the literature. The clear and succinct figures were arguably the strength of the paper, but the manuscript needs to do a better job describing the patterns portrayed and their importance for application. In general, we expected a greater discussion on the more broad implications of the study. Specifically, the discussion would greatly benefit from directly connecting the findings to possible ecological impacts and marine resource management given the inferences toward range shifts in the abstract and introduction.

Specific Comments

Lines 30-32: “predictability differences exist among species differing in temperature sensitivity of hypoxia vulnerability, especially along the northeast coast.” – there is a lot of language on differences here but not direction. It would be helpful to the reader to have information about location, magnitude, and direction of these differences.

Line 47: Start of sentence confusing as it sounds like O2 levels are already inadequate, consider rephrasing. Provide some examples of essential ecological activities as well to make this a more compelling statement.

Line 59: Explanation of terms is confusing. Makes A0 seem like 3 different things on first read. What are

Es and KB? I would move information from line 377 to 382 to this section to give more context

Line 63: Change “diverse” to “a diversity of”. Also, “Observed marine population distributions” – this is a bit confusing. Do you mean the distribution of population abundances or marine species distributions? You talk about occurrence in the following sentence so I would guess it’s just marine species distributions as population can be inferred to refer to abundances.

Line 101: What is the rationale for these depth ranges?

Line 107: Delete “decadal”, its redundant later in sentence

Line 110-11: “a substantial improvement” – please include specifics here. Subjective terms can be difficult for the reader to interpret.

Line 113: “Distinct differences in predictability” – same comment, please give specifics!

Line 133: “However, the medium- and high- E species are habitable” – do you mean have habitat? I don’t understand how the species are habitable.

Line 136: What? Get rid of the parentheses and describe both phenomena with respect to E_0 values, it’s confusing the way it’s currently written

Line 141-2: “viable habitats suggest shoaling or deepening” – sorry to beat this drum, but please focus on specificity. Where or what conditions lead to shoaling v. deepening?

Line 151: Why would they shoal if lower E_0 species have greater less sensitivity?

Line 152: Change “shrank” to “shrink”

Line 159: Would be helpful to have simple explanation of what ACC and NMAE actually measure/mean

Line 174: “significantly higher prediction skills” – again, same point. This language is persisting where it’s unclear to the reader what this means. E.g. ACC increases from .6 to .8? I’m going to stop listing these because they continue numerous times throughout the MS. Please avoid adjectives such as significant or distinct without including the parameters or statistics in-line.

Line 250-1: Add “of” after “rest”, “The rest LMEs have less differences, especially the lower depth layer of the southwest coast LMEs” – same comments as above...

Line 278: “ranging from 0.1 to 0.6 eV has notable higher prediction skills, and these differences are mainly contributed by the O_2 component” – range is good, but “higher prediction skills” and “mainly contributed” are vague!

Line 286: “Different from previous habitat models that solely relying” -> solely rely

Line 347: Awkward sentence, rephrase

Line 396: delete", and it has" and replace with "with"

Line 449: This is confusing. Do you mean "the maximum lead time between 1 and 10 years?"

Line 638: Remove "s" from "habitats"

Reviewer #4 (Remarks to the Author):

Thank you for the opportunity to review this manuscript. My specialty lies in use of the metabolic index rather than general oceanography/modelling and so my comments focus on that aspect of the manuscript for which I have no major concerns. See below for line comments. I am not qualified unfortunately to receive the ocean modelling components of the manuscript.

Line 59: terms hypoxia "tolerance" and "vulnerability" used to describe the metabolic index where the same experimental data is used to quantify both.

Figure 2a – since habitat is only viable at $\phi = 1$ could the colour bar represent that – i.e. it is hard to distinguish where habitat becomes viable with current colour scheme including zero

Line 135: does this pattern not reflect the cold temperatures in the north and high temperature sensitivity of hypoxia tolerance extending into these colder temperatures resulting in a predicted low p_{crit} and hence high ϕ ?

Line 137 – 139: I don't follow this logic – if habitat is limited on the south due to high temperatures and low oxy as previously stated then would more warming/deoxygenation not further contract habitat in the south? This needs to be explained more clearly.

Figure 2g-h – what do the codes of colours correspond to? Coding error?

Line 244: but different A_o was not tested in this manuscript?

We appreciate the reviewers' careful and constructive comments and suggestions on the manuscript. Our responses below are in blue font. Line numbers mentioned in the responses are referred to the track changes version.

Reviewer #1 (Remarks to the Author):

The manuscript "Skillful Multiyear Prediction of Marine Habitat Shifts Jointly Constrained by Ocean Temperature and Dissolved Oxygen" by Chen et al examines the potential predictability of species ecotypes using a physiologically-rooted approach.

This manuscript represents a major advance in our approach to ecological prediction in the ocean. Marine science has typically had very narrow focus on the environmental variables that shape the distribution of species, typically using temperature as the sole (or major) determinant of the ecological niche, and therefore to their prediction. This paper breaks with that tradition by highlighting the potential that oxygen can play in deriving predictions. It therefore represents a substantial breakthrough in the field. The paper is well written, convincing and without any major errors.

Thank you very much for your favorable comments!

If, and only if, another revision of the manuscript is required, I would encourage a few minor clarifications that would improve the quality of the paper

- The author's use of the word "species" is a little too loose. The re-prediction exercise is not based on the distribution of an actual species, but on some "representative" parameter values that the authors have chosen. This is fine (and indeed eminently sensible!) but I would prefer that this point is made clearer. The author's are clearly aware of it, and suggest the use of the phrase "ecotype" on line 380 – this is a good alternative that could be more widely adopted throughout the manuscript, including the abstract.

Thanks. The phrase "ecotype" has been widely adopted in the main text including the abstract. We have also moved the clarification of "ecotype" (on previous line 380) to the main text (lines 149-151).

- Similarly, it is important to note that this is an exercise in "potential predictability", as the "reference" dataset against which the predictions are compared is ultimately a product derived from the same model system. I understand why the author's have taken this approach – oxygen observations are rare – but this important detail is unfortunately buried in the methods. Again, I would like to see the authors be upfront about this, and adopt the language throughout the manuscript, including the abstract. A discussion of the impact of other oxygen products could be a useful addition to the discussion.

We thank the reviewer for this suggestion. We have now clarified the "potential predictability" issue (due to temporally and spatially sparse O₂ observations) in the Introduction section (lines 128-132), instead of stating this key info in the Methods section. We also have addressed and discussed this issue in the Discussion and Conclusion section (lines 696-700) - "However,

reliable observations or observation-based products of O₂ with spatio-temporal variability are urgently needed to be able to truly verify the realized prediction skills and move from potential predictability to verified prediction skill. These observational products exist for T, S, and chlorophyll, but are just beginning to be developed for O₂ (e.g., the GOBAI-O2 product) and other biogeochemical variables”.

- Finally, I am missing the link from the “ecotypes” back to the real species and their distribution, and ultimately to real-world predictions of species distributions. Given the decadal-time scales considered here, climate change is not really an important motivating factor – the work is more relevant to generating ecological forecast products for use in decision making. There is fortunately plenty of good work on this topic to cite e.g. Tommasi et al 2017, Payne et al 2017 and O’Kane et al 2023.

Long story short – this is a great article that should be published.

Mark R. Payne
Danish Meteorological Institute (DMI)
Copenhagen, Denmark

Thanks for these suggestions. We have added more species info - common names of species, to the corresponding traits distribution (A_c vs. E_o) in the **Supplementary Figure 1** and its caption (and for the three representative species in the main text - lines 151-153 and 542-544), in order to enhance the link between “ecotypes” and “real species”. Readers now can have a more direct feeling about which real species the low, medium, and high E_o ecotypes represent.

Correct, and thanks for the suggestion. We have added citations of the abovementioned important papers about marine ecological forecasting in the Conclusion and Discussion section (lines 497-500).

References:

Tommasi et al 2017 <https://www.sciencedirect.com/science/article/abs/pii/S0079661116301586>
Payne et al 2017 <https://www.frontiersin.org/articles/10.3389/fmars.2017.00289/full>
O’Kane et al 2023 doi: 10.3389/fclim.2023.1121626

Reviewer #2 (Remarks to the Author):

Co - reviewed with Reviewer #3

Reviewer #3 (Remarks to the Author):

Broad overview

We appreciated the opportunity to review a “Skillful Multiyear Prediction of Marine Habitat Shifts Jointly Constrained by Ocean Temperature and Dissolved Oxygen.” The manuscript validates a critical time scale of predictability from interannual to decadal scales – using a full suite of models to look beyond temperature to oxygen, and combining the two to address

organismal requirements. The premise of the study is worthy of publication but the manuscript itself should be revised further before acceptance. Our biggest concerns focus on the precision of the writing. Specifically, we are not getting information on the results of differences that allow the reader to assess whether the findings are significant or not and instead have to trust on the author's expertise. It's a pervasive issue throughout the manuscript but an easy topic to address!

For the methods, a table outlining the equations and variables that make them up would be more useful than explaining everything in the main body of the text and would reduce overall complexity and clutter. This would also provide a one stop location to quickly review terms while interpreting results. We suggest avoiding the use of acronyms as much as possible and instead spelling out names if something is only mentioned a couple times as there are many abbreviations to keep track of. We appreciated the section starting on L203 which digs into the mechanistic drivers of predictability as it helps the manuscript be of broad interest to the journal's readership. Overall however, we had a hard time digging into the take-home messages because of the lack of specificity in writing but feel that if the writing were more clear, the manuscript could be a good and important contribution to the literature. The clear and succinct figures were arguably the strength of the paper, but the manuscript needs to do a better job describing the patterns portrayed and their importance for application. In general, we expected a greater discussion on the more broad implications of the study. Specifically, the discussion would greatly benefit from directly connecting the findings to possible ecological impacts and marine resource management given the inferences toward range shifts in the abstract and introduction.

We thank the reviewer for the constructive comments and suggestions. We have improved the precision of writing not only in places suggested by the reviewers (see responses below), but throughout all four Results subsections with more details (e.g., numbers, regions, and traits) added describing the key results (e.g., patterns shown in the figures). Specificity has been largely improved in writing, as well as the take-home messages have been made more clear in the Results and Conclusion and Discussion sections (see the track changes). For example, we largely revised the paragraphs in the last Results subsection - we first describe the major predictability differences across different E_0 ecotypes in the first two paragraphs, then the mechanistic controls to the differences in the following paragraphs. We deliver the main messages in the first two sentences of each paragraph followed by the examples and explanations with detailed numbers, regions, and ecotypes included.

For the methods, we have moved the definition equations (e.g., Equation 1) and associated explanation of parameters to the Methods section (The Metabolic Index) as a one stop location, instead of showing the equation in the main text and explaining some parameters in the Methods section. So, readers only need to go over the Method section to understand the index and review all the terms.

We have also spelled out some less used names as suggested, instead of using the acronyms, e.g., the Gulf of Mexico, Aleutian Islands, Gulf of Alaska, California Current, Gulf of California, and Scotian Shelf. We only keep the acronyms for some long and frequently used names, e.g., Insular Pacific Hawaiian, Northeast U.S. Shelf, Southeast U.S. Shelf, Eastern Bering Sea, and Labrador-Newfoundland.

We have also added more discussions in the Discussion and Conclusion section on the implications of the study and importance for application (see the track changes), by directly connecting the findings to possible ecological impacts and marine resource management (lines 664-677).

Specific Comments

Lines 30-32: “predictability differences exist among species differing in temperature sensitivity of hypoxia vulnerability, especially along the northeast coast.” – there is a lot of language on differences here but not direction. It would be helpful to the reader to have information about location, magnitude, and direction of these differences.

Due to the words limit of the Abstract, we only provide with a short take-home message (marine species with different temperature sensitivities have notable predictability differences in the same LME using the CESM-DPLE system, especially in the northeast coast LMEs). We have indicated the location that have the most notable predictability difference (the northeast coast LMEs), and added magnitude in predictability timescale difference (ranging from 2 to 10 years) in the sentence (lines 32-33).

Line 47: Start of sentence confusing as it sounds like O₂ levels are already inadequate, consider rephrasing. Provide some examples of essential ecological activities as well to make this a more compelling statement.

We have rephrased the sentence by removing “present-day”, suggesting O₂ may be inadequate under warmer conditions (line 54). We have also added essential ecological activities “such as feeding, defense, and reproduction” to make it a more compelling statement (line 53).

Line 59: Explanation of terms is confusing. Makes A₀ seem like 3 different things on first read. What are E_s and K_B? I would move information from line 377 to 382 to this section to give more context

We have moved the Equation (1) and its associated explanation of parameters (A₀, E₀, K_B, etc.) to the Method section (The Metabolic Index) instead of showing the equation in the main text and explaining some parameters in the Methods, so that readers only need to go over the Method section as a one stop location to quickly review all terms.

Line 63: Change “diverse” to “a diversity of”. Also, “Observed marine population distributions” – this is a bit confusing. Do you mean the distribution of population abundances or marine species distributions? You talk about occurrence in the following sentence so I would guess it’s just marine species distributions as population can be inferred to refer to abundances.

Done.

Yes, it’s marine species distributions. We have changed the word “marine population distributions” to “marine species distributions” as suggested.

Line 101: What is the rationale for these depth ranges?

We have added rationale for these two depth ranges in the main text - "...in two depth habitats (0-200 m, the surface layer or epipelagic zone where most of the visible lights exist, and 200-600 m, the thermocline layer within the mesopelagic zone or twilight zone) of the upper ocean ..."

(lines 121-122).

Line 107: Delete "decadal", its redundant later in sentence

Done.

Line 110-11: "a substantial improvement" – please include specifics here. Subjective terms can be difficult for the reader to interpret.

Done. We have added specifics here – "showing a substantial improvement in predictability timescale from 2 to 6 years against a simple persistence forecast".

Line 113: "Distinct differences in predictability" – same comment, please give specifics!

Done. We have added specifics here – "Distinct differences in predictability exist among ecotypes with different E_o traits in the LMEs, especially along the northeast coast with the average predictability timescales ranging from 2 to 10 years for low- E_o species."

Line 133: "However, the medium- and high- E species are habitable" – do you mean have habitat? I don't understand how the species are habitable.

Yes. We have changed the words "are habitable" to "inhabit" and corrected other similar errors in the manuscript (lines 157, 164).

Line 136: What? Get rid of the parentheses and describe both phenomena with respect to E_0 values, it's confusing the way it's currently written

Done. We have rephrased the sentence as suggested (lines 169-173).

Line 141-2: "viable habitats suggest shoaling or deepening" – sorry to beat this drum, but please focus on specificity. Where or what conditions lead to shoaling v. deepening?

Done. We have modified the sentence as suggested (lines 174-176) by adding "Within all these LMEs" (where) and "in response to the interannual variations of environmental O_2 and T" (what conditions).

Line 151: Why would they shoal if lower E_0 species have greater less sensitivity?

Lower- E_o means they have relatively smaller temperature sensitivity, which suggests less metabolic demand changes arising from environmental T changes (the denominator of the index

defined in Equation 1). However, O_2 availability in the form of p_{O_2} (the numerator of the ratio) is still affected by variables of T, S, and O_2 concentration. In other words, large decrease of the O_2 availability (in the form of p_{O_2}) due to warming or deoxygenation could still lead to shoaling of viable habitats even for less temperature sensitivity (E_o) species. We have added a sentence in the text to clarify this point (lines 222-224).

Line 152: Change “shrank” to “shrink”

Done.

Line 159: Would be helpful to have simple explanation of what ACC and NMAE actually measure/mean

Done. We have added simple explanations in the text of the second Result subsection (lines 230-234).

Line 174: “significantly higher prediction skills” – again, same point. This language is persisting where it’s unclear to the reader what this means. E.g. ACC increases from .6 to .8? I’m going to stop listing these because they continue numerous times throughout the MS. Please avoid adjectives such as significant or distinct without including the parameters or statistics in-line.

In fact, the words “significant” or “significantly” mean the ACCs are significant at the 95% confidence level or the improvement of ACCs are significant at the 95% confidence level (via significance tests; see the Methods section for prediction skill metrics). So, we have added clarifications and other specifics in the manuscript when first using the adjectives “significant” and “distinctive”, to avoid misleading readers (lines 239; 275). We have also added specific values of ACC increases in the text when using the words, as suggested by the reviewer (lines 276-279; 294-299; 301-302).

Line 250-1: Add “of” after “rest”, “The rest LMEs have less differences, especially the lower depth layer of the southwest coast LMEs” – same comments as above...

Done.

We have added specific differences in predictability across different ecotypes in these rewritten paragraphs (lines 406-447).

Line 278: “ranging from 0.1 to 0.6 eV has notable higher prediction skills, and these differences are mainly contributed by the O_2 component” – range is good, but “higher prediction skills” and “mainly contributed” are vague!

We have added specifics to avoid vague statement in the revised paragraph (lines 452-480).

Line 286: “Different from previous habitat models that solely relying” -> solely rely

Done.

Line 347: Awkward sentence, rephrase

Done. We have rephrased this sentence (lines 666-667).

Line 396: delete”, and it has” and replace with “with”

Done.

Line 449: This is confusing. Do you mean “the maximum lead time between 1 and 10 years?”

No. We have rephrased this sentence to make it clear (lines 820-821). We have also provided and modified the example for the definition following the sentence (lines 821-823).

Line 638: Remove “s” from “habitats”

Done.

Reviewer #4 (Remarks to the Author):

Thank you for the opportunity to review this manuscript. My specialty lies in use of the metabolic index rather than general oceanography/modelling and so my comments focus on that aspect of the manuscript for which I have no major concerns. See below for line comments. I am not qualified unfortunately to review the ocean modelling components of the manuscript.

We thank the reviewer for the constructive comments on the Metabolic Index.

Line 59: terms hypoxia “tolerance” and “vulnerability” used to describe the metabolic index where the same experimental data is used to quantify both.

Yes. The “temperature sensitivity of hypoxia vulnerability” (E_o) and “hypoxia tolerance” (A_o) are synchronously derived from the same published laboratory data, based on their definitions for the Metabolic Index [*Deutsch et al.*, 2015&2020]. We have added this info in the text (lines 64-66).

Figure 2a – since habitat is only viable at $\phi = 1$ could the colour bar represent that – i.e. it is hard to distinguish where habitat becomes viable with current colour scheme including zero

Thanks for the suggestion. We have added a red line on the colorbar of Figure 2 to indicate where $\phi = 1$. In fact, the red contours in Figure 2(a), (c), and (e) have suggested $\phi = 1$, as described in the figure caption. The color scheme has to include zero as some regions are almost unviable with the ϕ approaching zero.

Line 135: does this pattern not reflect the cold temperatures in the north and high temperature

sensitivity of hypoxia tolerance extending into these colder temperatures resulting in a predicted low p_{crit} and hence high ϕ ?

Yes, the spatial patterns for medium- and high- E_o species has reflected the lower $p_{crit_{O_2}}$ required in low-T regions, thus leading to higher ϕ (ϕ) values. We have incorporated this info in the text (lines 165-166).

Line 137 – 139: I don't follow this logic – if habitat is limited on the south due to high temperatures and low oxy as previously stated then would more warming/deoxygenation not further contract habitat in the south? This needs to be explained more clearly.

Yes, warming or deoxygenation would contract habitats northward by retreating the southern boundaries of habitats, and cooling or oxygenation would expand habitats southward by adding habitat, therefore habitat shifts are in the meridional (north-south) direction. We have rephrased this sentence in the text (lines 169-173).

Figure 2g-h – what do the codes of colours correspond to? Coding error?

The colors in Figure 2h represent different LMEs within a sub-group (e.g., Northwest Coast LMEs), as indicated in the legend of each panel in Figure 2g. They are not coding error. We have added two sentences in the figure caption to clarify this point (lines 1063-1064, 1070-1071).

Line 244: but different A_o was not tested in this manuscript?

Yes, this is because the different- A_o (actually A_c), same- E_o marine species (or 'ecotypes') have exactly the same predictability skills in ACC (Anomaly Correlation Coefficient) due to the linear scaling effect of the A_c trait on the habitat viability (ϕ). We have explained it in lines 144-149 and already clarified it in the Discussion and Conclusion section (lines 656-663) - "We focus on habitat shifts and viability predictions for different E_o ecotypes with the same A_c (10 atm^{-1}) trait. Due to the linear scaling effect of the A_c trait, these results are applicable to any other ecotypes with the same E_o but different A_c traits. For example, the interannual habitat shifts are similar for ecotypes with the same E_o but smaller (larger) A_c traits, only having the habitat boundaries moving northward (southward) or towards high (low) ϕ regions. The prediction skill assessments in ACC are also the same between the same- E_o different- A_c ecotypes, as the linear scaling coefficient A_c is self-canceled."

REVIEWERS' COMMENTS

Reviewer #2 (Remarks to the Author):

Co - reviewed with Reviewer #3

Reviewer #3 (Remarks to the Author):

The incorporated revisions definitely improved the manuscript, particularly with regards to specificity and connecting to broader ecological research. Just a few small typos to correct, otherwise I think this is ready to go!

Line 198: Change "stem" to "stems"

Line 313: Change to "The S component also presents as an important contributor...."

Reviewer #4 (Remarks to the Author):

Thank you for addressing my limited queries related to the metabolic index. I am satisfied with the responses. Thank you for the opportunity to review the manuscript and I endorse its publication. Noting again that I am not qualified to assess the ocean modelling component of the manuscript that forms a large portion.

Reviewer #2 (Remarks to the Author):

Co - reviewed with Reviewer #3

Reviewer #3 (Remarks to the Author):

The incorporated revisions definitely improved the manuscript, particularly with regards to specificity and connecting to broader ecological research. Just a few small typos to correct, otherwise I think this is ready to go!

We appreciate the reviewers' careful and constructive comments and suggestions on the manuscript.

Line 198: Change “stem” to “stems”

Done.

Line 313: Change to “The S component also presents as an important contributor....”

Done.

Reviewer #4 (Remarks to the Author):

Thank you for addressing my limited queries related to the metabolic index. I am satisfied with the responses. Thank you for the opportunity to review the manuscript and I endorse its publication. Noting again that I am not qualified to assess the ocean modelling component of the manuscript that forms a large portion.

We thank the reviewer again for the constructive comments on the Metabolic Index.